# Decadal oscillation provides skillful multiyear predictions of Antarctic sea ice

Yusen Liu[1], Cheng Sun ®[1] ✉, Jianping Li ®[2,3], Fred Kucharski ®[4], Emanuele Di Lorenzo[5], Muhammad Adnan Abid ®[4,6] & Xichen Li ®[7]

Over the satellite era, Antarctic sea ice exhibited an overall long-term increasing trend, contrary to the Arctic reduction under global warming. However, the drastic decline of Antarctic sea ice in 2014–2018 raises questions about its interannual and decadal-scale variabilities, which are poorly understood and predicted. Here, we identify an Antarctic sea ice decadal oscillation, exhibiting a quasi-period of 8–16 years, that is anticorrelated with the Pacific Quasi-Decadal Oscillation ($r = -0.90$). By combining observations, Coupled Model Intercomparison Project historical simulations, and pacemaker climate model experiments, we find evidence that the synchrony between the sea ice decadal oscillation and Pacific Quasi-Decadal Oscillation is linked to atmospheric poleward-propagating Rossby wave trains excited by heating in the central tropical Pacific. These waves weaken the Amundsen Sea Low, melting sea ice due to enhanced shortwave radiation and warm advection. A Pacific Quasi-Decadal Oscillation-based regression model shows that this tropical-polar teleconnection carries multi-year predictability.

Antarctic sea ice plays a crucial role in the exchange of heat, momentum, and water masses, modulating global atmospheric and oceanic circulations[1–5]. It is characterized by complex variations on a variety of timescales superimposed on an overall long-term increasing trend during the satellite era[6–9]. Despite this, the quick retreat of Antarctic sea ice from 2014 to 2018 has garnered a lot of attention[3,10–13]. These long-term fluctuations have been connected to dynamic processes in the Southern Ocean[14–17] and large-scale climate modes in the Pacific and Atlantic Oceans[18–23]. For example, a previous study suggested that the increasing trend of Antarctic sea ice in the 2000s is driven by the Interdecadal Pacific Oscillation (IPO)[18], which shows drastic multidecadal fluctuations (20–30 years). On an interannual period (2–8 years), the Southern Annular Mode (SAM) and a circumpolar zonal wave-3 (ZW3) are primary local circulation patterns that affect the variability of sea ice through wind-driven

dynamic and thermodynamic processes[24–26]. Remotely, the El Niño-Southern Oscillation (ENSO)-related tropical Pacific SST warming causes contrasting changes in sea ice concentration (SIC) over the Weddell and Ross Seas, referred to as the Antarctic sea ice dipole[27–29]. Uncovering the complex variations of Antarctic sea ice is a basis for advancing its prediction. However, the quasi-decadal (8–16 years) variability and the recent drastic fluctuations in sea ice have not been fully understood, and accurate prediction of Antarctic sea ice remains challenging[30]. The interannual sea ice predictability only reaches three years using intermediate complexity climate models[31,32]. Meanwhile, the decadal prediction of Antarctic sea ice is far from skillful since most CMIP5 models are unable to reproduce the observed increasing trend[33,34]. A recent study demonstrated that the predicting skill of sea ice over the Ross Sea on the decadal scale is still very limited in spite of refined model initialization[30]. Thus, there

[1]State Key Laboratory of Remote Sensing Science, Faculty of Geographical Science, Beijing Normal University, Beijing, China. [2]Frontiers Science Center for Deep Ocean Multi-spheres and Earth System (DOMES)/Key Laboratory of Physical Oceanography/Academy of Future Ocean/College of Oceanic and Atmospheric Sciences, Ocean University of China, Qingdao, China. [3]Laoshan Laboratory, Qingdao, China. [4]The Abdus Salam International Centre for Theoretical Physics, Trieste, Italy. [5]Department of Earth, Environmental, and Planetary Sciences, Brown University, Providence, RI, USA. [6]Atmospheric, Oceanic and Planetary Physics, Department of Physics, University of Oxford, Oxford, UK. [7]Institute of Atmospheric Physics, Chinese Academy of Sciences, Beijing, China. ✉e-mail: scheng@bnu.edu.cn

is an urgent need to improve the Antarctic sea ice predictability on the decadal scale.

Low-frequency oscillatory phenomena have been observed in the Pacific basin on a variety of temporal scales[35], providing an important source of predictability[36]. Other than the interannual and multidecadal modes, the quasi-decadal oscillating signals in the Pacific are less understood and more pronounced on the regional scale[35,37–39]. One of these decadal modes is located over the central tropical Pacific, where the sea surface temperature (SST) exhibits alternative warming and cooling phases, with a quasi-period of 8–16-yr, referred to as the Pacific Quasi-Decadal Oscillation (PQDO)[37,39–41]. Previous studies regarded the PQDO as a non-linear component of ENSO[42], while other studies also suggested that the dynamical coupling between the tropical SST heating and extratropical atmospheric forcing associated with the Kuroshio Extension plays a role[43–47]. It has been conclusively proved that the PQDO is an independent quasi-periodic mode and fundamentally different from ENSO[42]. Also, the PQDO can be differentiated from the IPO in terms of their temporal scales[48]. Moreover, the PQDO has profound influences on regional climate variability on the quasi-decadal time scale, such as tropical cyclones and inland precipitation[49–52]. Investigating the climate impacts of the PQDO would enlighten our understanding of quasi-decadal climate variations and improve climate predictability.

In this study, we investigate the decadal variability in Antarctic sea ice and examine its quasi-periodic nature. The underlying mechanism is inspected in historical Coupled Model Intercomparison Project Phase 6 (CMIP6) simulations and a dedicated pacemaker experiment conducted with the ICTPAGCM-NEMO coupled model. A PQDO-based sea ice regression model is further constructed to estimate the SIC changes in the next decade.

## Results

### Decadal oscillation in Antarctic sea ice

To identify the predominant mode of Antarctic sea ice variability, we employ the Empirical Orthogonal Function (EOF) on the annual mean SIC (within 50°S–90°S) derived from the NSIDC dataset for the period 1981–2020. The leading mode of EOF explains about 16.9% of the variance (Supplementary Fig. 1a). The corresponding principle component (PC) shows an overall increasing trend in SIC (Supplementary Fig. 1b). The SIC increases over the Ross[53] and Weddell Seas while decreasing over the Amundson and Bellingshausen Seas, which corresponds well with the observed trend pattern (Supplementary Fig. 1c). The second mode of EOF highlights an oscillating feature, with the maximum loading positioned over the Ross-Amundsen Seas (Fig. 1a). It explains 13.1% total variance (Supplementary Fig. 2a) and 26.3% decadal variance (Supplementary Fig. 2b) of the entire Antarctic SIC, indicating that the PC2 is a dominant decadal mode of Antarctic sea ice. Note that the EOFs conducted on monthly SIC data show consistent temporal and spatial characteristics (not shown here) and can be well separated following the "rule of thumb" in ref. 54 (see Methods). The Ross-Amundsen Seas SIC strongly coincides with the PC2 ($r = 0.88$), consistently showing a quasi-period of 8–16 years and a spectral peak at roughly 12–14 years (Fig. 1b, c). Furthermore, we compute the EOFs using detrended SIC data, where the PC1 strongly correlates with the PC2 from the raw data ($r = 0.77$), explaining 19% variance of Antarctic SIC and exhibiting a spectral peak at 12–14 years (Supplementary Fig. 2c, d). It demonstrates that the decadal oscillation is one of the primary modes of Antarctic sea ice variability, which is most pronounced in the Ross-Amundsen Seas, but also has implications for the decadal oscillatory features elsewhere in the Antarctic Seas. We then apply an 8-yr lowpass filter to the Ross-Amundsen Seas SIC and the PC2 to extract the decadal component of the SIC variability (Fig. 1d). Antarctic sea ice has gone through more than two cycles in the past forty years, with three positive phases (1984–1990; 1996–2002; 2008–2012) and two negative phases in between. The decline of the Ross-Amundsen Seas SIC from 2012 and reclimb from 2018 may be a part of the current cycle. The two series are largely in phase ($r = 0.96$), further suggesting that the Ross-Amundsen Seas SIC is representative of the decadal component of Antarctic sea ice

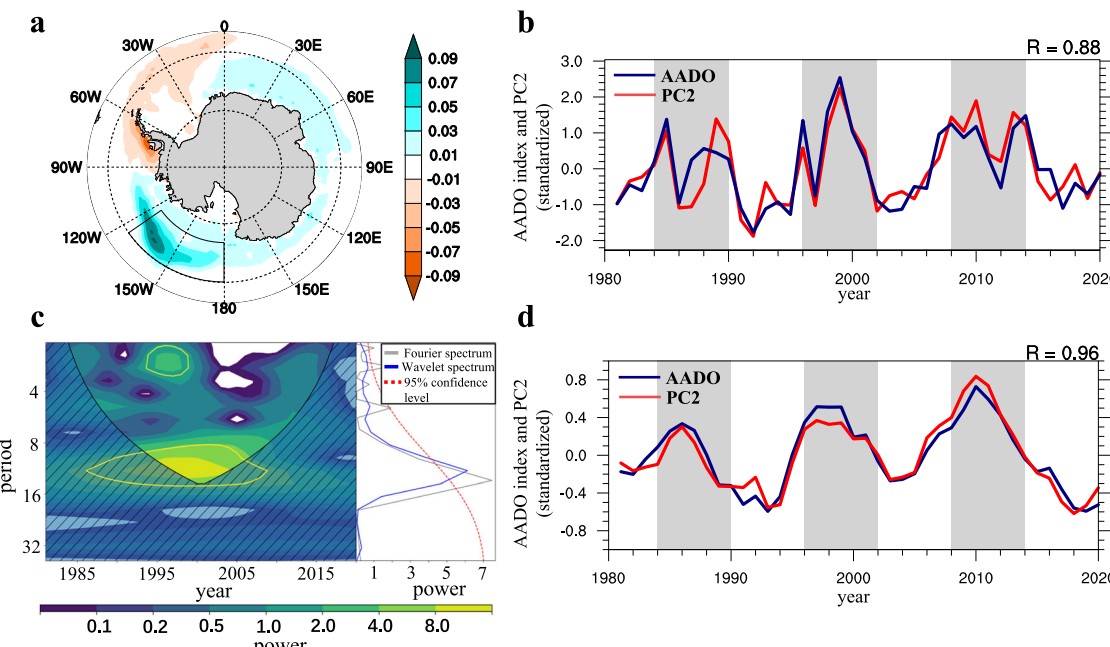

**Fig. 1 | Decadal oscillation in Antarctic sea ice concentration. a** The EOF2 pattern of Antarctic sea ice concentration (SIC). **b** The corresponding PC2 and the Antarctic decadal oscillation (AADO) index defined as the area-weighted average of SIC in the Ross-Amundsen Seas (180°–125°W, 60°S–75°S). **c** The local wavelet power spectrum using the Morlet wavelet, and the global wavelet/Fourier spectrums (right-hand panel) of the AADO index. The yellow contour indicates the 95% significance level using a red-noise background spectrum. The red dash line in the right-hand panel indicates the 95% confidence level for the global wavelet spectrum. **d** The 8-yr lowpass filtered series of the AADO index and the PC2 for the period 1981–2020. The gray shading indicates the positive phase of the AADO. The series has been detrended and normalized. Base map for Fig. 1a was generated using the NCAR Command Language (version 6.6.2).

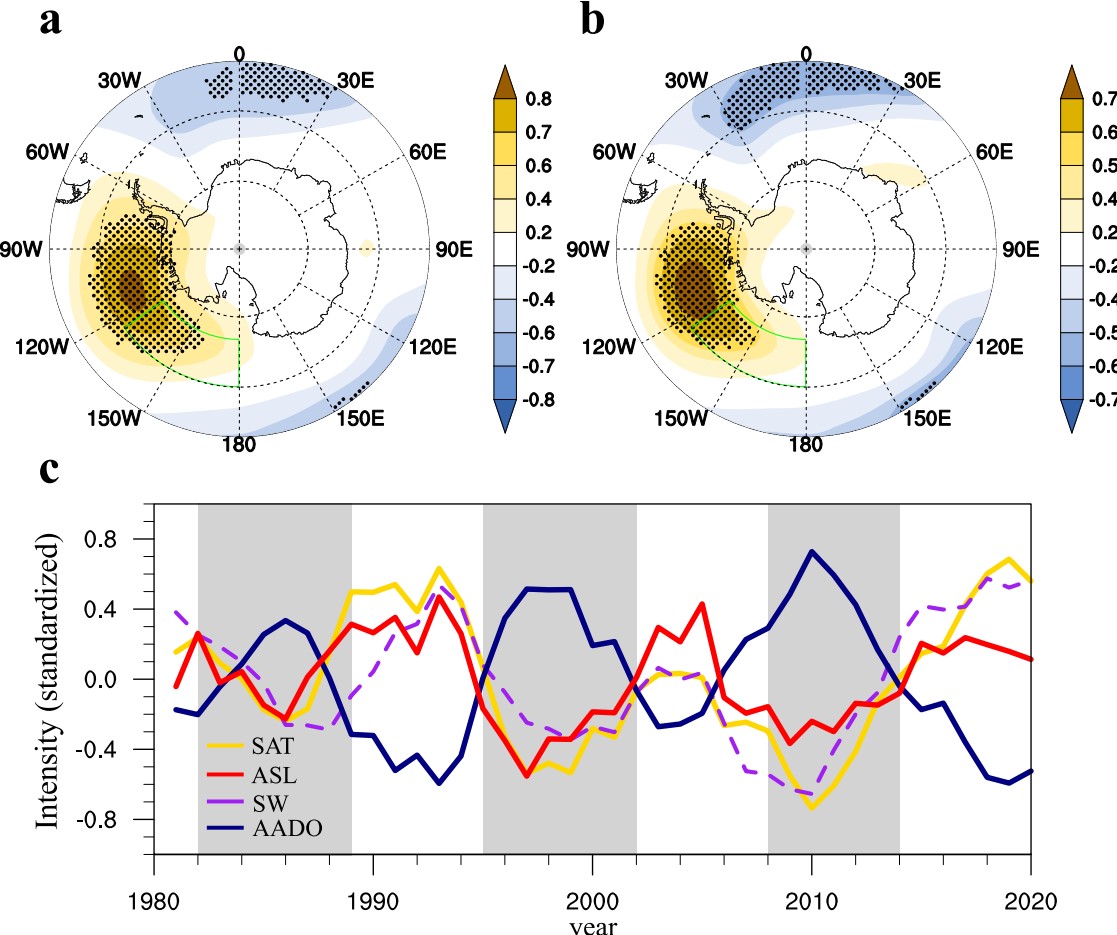

**Fig. 2 | Atmospheric forcing of the Antarctic decadal oscillation.** The correlation maps of (**a**) the Antarctic decadal oscillation (AADO) index (multiplied by −1) and (**b**) surface air temperature (SAT) averaged over the Ross-Amundsen Seas (180°–125°W, 60°S–75°S) with 200 hPa geopotential height. The dotted shading indicates the correlation coefficient is significant at the 95% confidence level. **c** The detrended and normalized time series of the AADO index (blue), the Amundsen Sea

Low (ASL) index (red), and domain-averaged SAT (yellow) and shortwave radiation (SW) (positive downward; dashed purple line) for the period 1981–2020. The time series has been preprocessed by an 8-yr low-pass filter in order to isolate the decadal variabilities. Base maps for Fig. 2a, b were generated using the NCAR Command Language (version 6.6.2).

variability, with a notable quasi-period of 8–16 years. We, therefore, refer to it as the Antarctic decadal oscillation (abbreviated AADO). The AADO index (see Index definitions in Methods) using the OISST and the HadISST SIC datasets (Supplementary Fig. 3) consistently shows a decadal oscillation. We may conclude that the AADO is a robust mode and independent of SIC datasets.

Antarctic sea ice is modulated by thermodynamic and dynamic processes largely driven by large-scale atmospheric circulations[3]. The local factors affecting sea ice display a consistent quasi-decadal oscillation (Fig. 2). The AADO strongly coincides with local surface air temperature (SAT) ($r = -0.97$, lowpass filtered). The lead-lag correlation between SAT and SIC indicates that the SIC melting driven by preceding and contemporary SAT warming is stronger than its feedback to maintain the anomalous SAT (Supplementary Fig. 4a). Thus, the local SAT plays an essential triggering role causing sea ice to melt. The correlation maps of the AADO index (reversed) and the area-averaged SAT over the Ross-Amundsen Seas with 200 hPa geopotential height consistently show high anomalies, which agree well with the climatological Amundsen Sea Low (ASL) (Figs. 2a, b). The weakened ASL enhances shortwave radiation (Supplementary Fig. 4b) and drives anomalous warm air advection (Supplementary Fig. 4c) to the west, causing regional SAT warming ($r = 0.84$) and, consequently, SIC decline ($r = -0.87$). We evaluate the relative contributions of shortwave radiation and temperature advection to the SIC changes using

standardized partial regression coefficients (−0.83 and −0.13, respectively) from a multivariate regression model of the AADO. It suggests that the radiative heating term may play a more important role. Based on the above analysis, the ASL is an essential local atmospheric forcing responsible for the changes in SIC via the thermodynamic process. The ASL exhibits a consistent 8–16-yr quasi-period through wavelet analysis (Supplementary Fig. 5) as that in the AADO. In Fig. 2c, the ASL shows an anticorrelated relationship with the AADO, together with synchronized decadal fluctuations in SAT and shortwave radiation.

Furthermore, we inspect the temporal coherence of the AADO with subsurface temperature and SIC advection (see Methods). Both subsurface temperature[19] and sea ice advection may influence the low-frequency variability of Ross-Amundsen Seas sea ice ($r = -0.52$ and 0.11, respectively). However, the quasi-periodicity found in the AADO is obscure in those two variables (Supplementary Fig. 4d, e), which are unlikely the driving factors of such a decadal oscillation. Meanwhile, SAT warming explains over 90% low-frequency variance of the Ross-Amundsen Seas SIC, with synchronized quasi-periods of 8–16 years. The analysis presented above reveals that the ASL and the associated thermodynamic processes are crucial to the development of the AADO. The quasi-periodicity on a scale of 8–16 years is consistently found in sea ice and the associated geopotential height and surface air temperature, further indicating that the AADO acts as a robust mode of variability in the Antarctic sea ice-atmosphere coupled system.

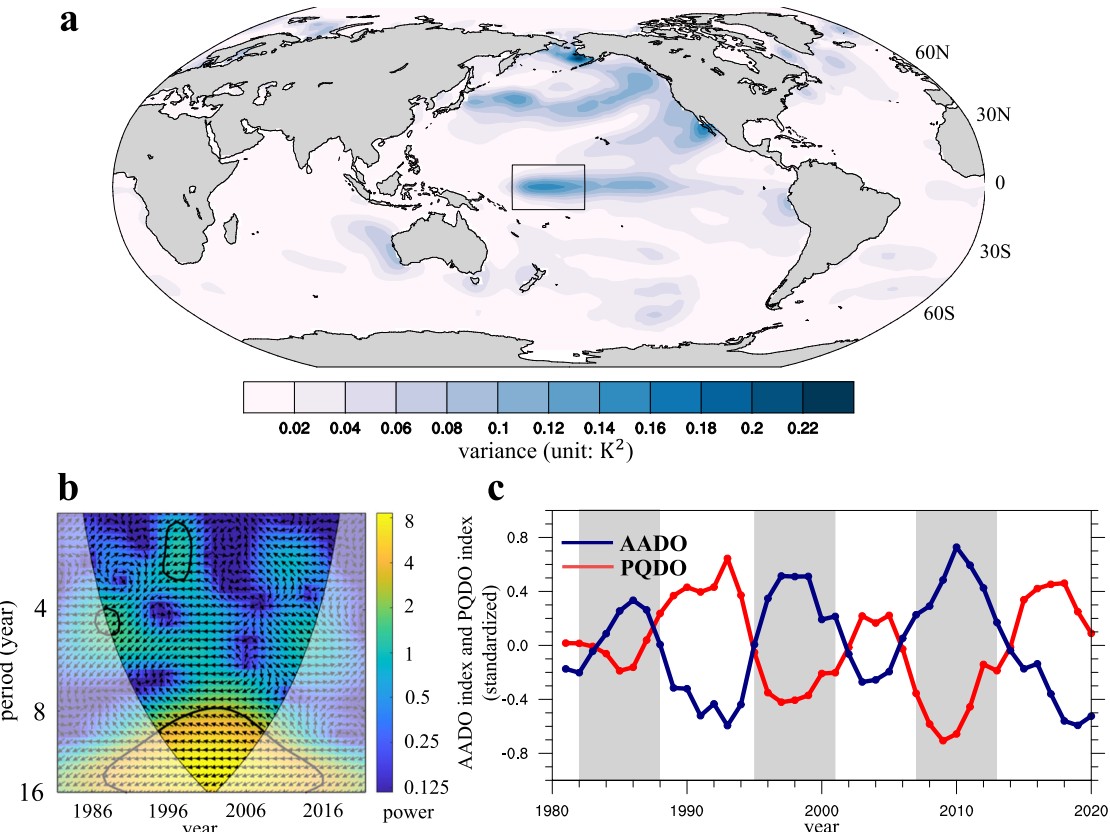

**Fig. 3 | Linking the Antarctic decadal oscillation to the Pacific Quasi-decadal oscillation. a** The sea surface temperature (SST) variance on the decadal scale (8-16-yr bandpass filtered, unit: K$^2$). **b** The cross-wavelet transform analysis between the Pacific quasi-decadal oscillation (PQDO) and the Antarctic decadal oscillation (AADO). The black contour indicates a 95% confidence level. The relative phase relationship is shown as arrows (with in-phase pointing right). **c** The 8-yr lowpass filtered time series of the PQDO index and the AADO index for the period 1981–2020 (detrended and normalized). Base map for Fig. 3a was generated using the NCAR Command Language (version 6.6.2).

## Linking the AADO to the PQDO

In this section, we attempt to explain the mechanisms behind the AADO by seeking signals that share a synchronized decadal fluctuation in the ocean. Here, we isolate the quasi-decadal component (8–16-yr bandpass filtered) of SST variance (Fig. 3a). The central tropical Pacific shows the most prominent signal, which is usually referred to as the PQDO[37]. The cross-wavelet between the PQDO index (see Methods) and the AADO index (Fig. 3b) indicates a compatible quasi-period of 8–16 years, with reversed phases. Further comparison of the two indices (8-yr low-pass filtered) shows synchronized decadal fluctuation (Fig. 3c) with a correlation coefficient of −0.90. Both the PQDO and the AADO experience approximately three complete cycles during the satellite observation, and the phase transitions between them are in good agreement. The above analysis reveals coherent quasi-periodicities, implying that the PQDO could be a potential driver of the AADO on the decadal scale.

The PQDO-AADO relationship and the underlying mechanisms are further inspected. In the observation, the PQDO-related SST warming intensifies tropical convection, accelerating the Hadley circulation and causing a disruption in the subtropical jet to excite a Rossby wave train that is propagating poleward[21]. Such a tropical-polar teleconnection pathway consists of alternating low/high pressure centers located over New Zealand and the Amundsen Sea (Fig. 4a), linking the PQDO to the atmospheric circulation anomalies over the Antarctic. The climatological ASL is severely suppressed by a high-pressure center at the downstream end of the wave train, which leads to surface warming from increased shortwave radiation and abnormally enhanced warm advection over the west flank. Consequently, the increased surface air temperature causes sea ice to melt (Fig. 4c) in response to the positive phase PQDO and the associated tropical-polar wave train. Note that the wave train pattern induced by the PQDO somewhat resembles the one excited by ENSO. However, the weakened ASL downstream the wave train exhibits considerable variation on the frequency band of 8–16 years, which cannot be explained by the interannual ENSO signal.

The central tropical Pacific pacemaker experiment (CP_EXP, see Methods) is conducted to inspect the direct response of Antarctic sea ice to the PQDO. The ICTP-NEMO coupled model has capability to reproduce the observed tropical-polar teleconnection and the decadal oscillation in the Ross-Amundsen Seas SIC (Supplementary Fig. 6). In the CP_EXP, we find a similar Rossby wave train pattern that propagates from the central tropical Pacific towards the polar region, with an anomalous high weakening the climatological ASL (Fig. 4b). The simulated SIC over the Ross Sea declines significantly in response to the PQDO (Fig. 4d). However, the simulated ASL deviates to the west, in accordance with the westward shifts in radiative heating and warm air advection, introducing biases in reproducing the SIC over the Amundsen Sea. Nevertheless, the coupled model successfully reproduces the ASL-related thermodynamic feedback between the local SAT and SIC ($r = -0.96$) and the anticorrelated relationship between the PQDO and AADO ($r = -0.53$) (Fig. 4e). The SIC in this PQDO-forced experiment shows a consistent quasi-periodicity on the decadal scale, with a spectral peak at approximately 12–14 years (Supplementary Fig. 7), as that in the observation. The pacemaker experiment isolates the role of PQDO in modulating the decadal oscillation of Antarctic sea ice, providing modeling evidence for the tropical-polar teleconnection pattern.

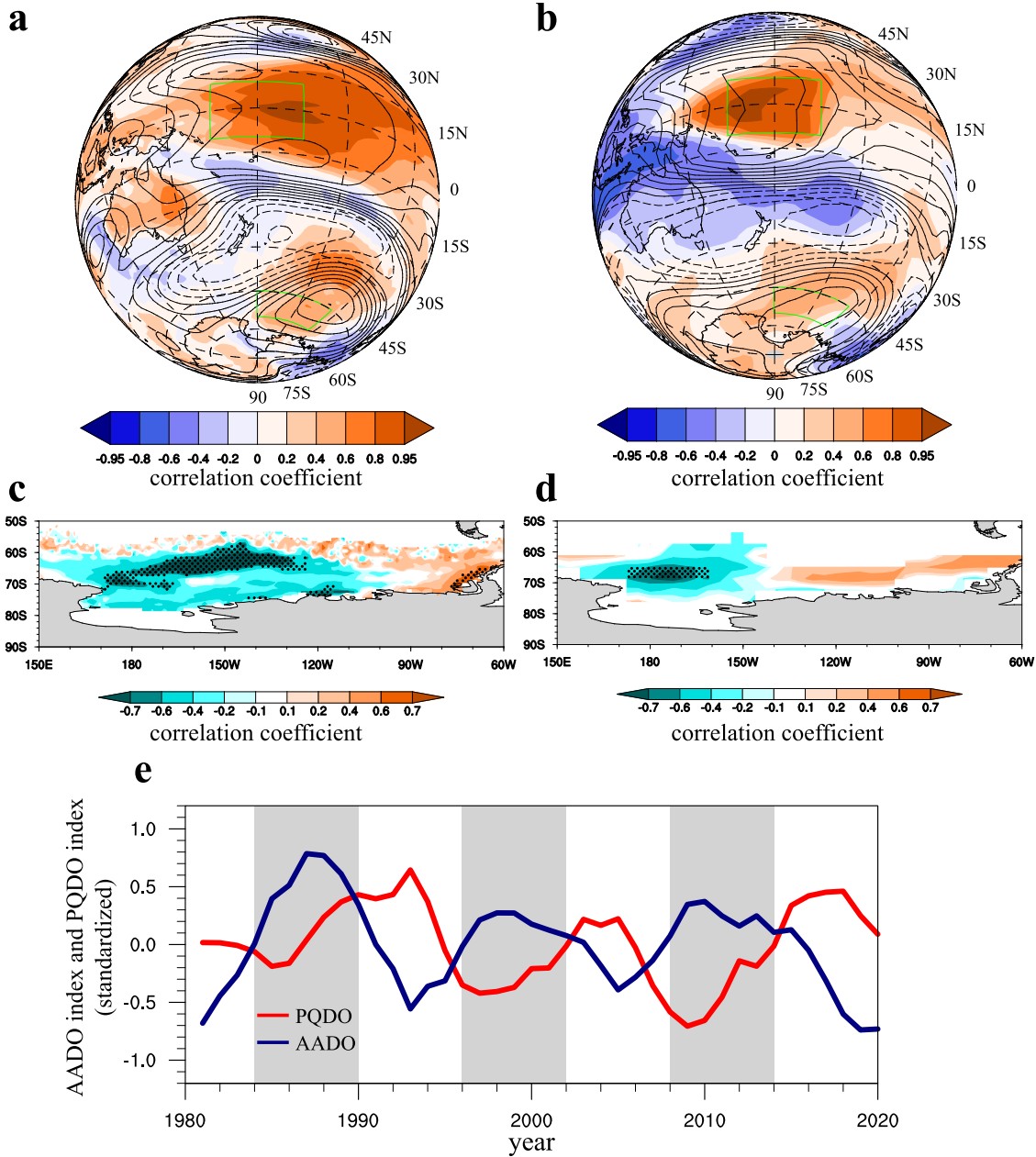

**Fig. 4 | Tropical-polar teleconnections.** The correlation maps of the Pacific-quasi-decadal oscillation (PQDO) index with 200 hPa geopotential height (contour) and surface air temperature (shading) in the (**a**) observation and (**b**) the central tropical Pacific pacemaker experiment (CP_EXP). **c**, **d** are the correlation maps of the PQDO index with Antarctic sea ice concentration in the observation and the CP_EXP, respectively, and have been preprocessed by 8-yr lowpass filtering. The dotted shading indicates the correlation coefficient is significant at the 95% confidence level. **e** The detrended and normalized time series of the PQDO index and the simulated Antarctic decadal oscillation (AADO) index in the CP_EXP. The model output is analysed from 1981 to 2020, consistent with the observation. Base maps for Fig. 4a–d were generated using the NCAR Command Language (version 6.6.2).

In CMIP6 historical simulations (Supplementary Table. 1), over half of the models replicate the central tropical Pacific SST's quasi-decadal variance ratio at over 10%. (Fig. 5a). We select 12 out of the 29 models, which show relatively good PQDO reproducibility with a variance ratio greater than 14%. Six of them also successfully reproduce the quasi-decadal signal in sea ice (Fig. 5b), and nine models indicate a significant quasi-decadal connection between the PQDO and the AADO, with negative correlation coefficients lower than −0.4 (Fig. 5c). The models with the more realistic intensity of SST variance and the quasi-decadal Ross-Amundsen Seas SIC variability simulate significant correlations between the PQDO and AADO index.

Here, we select four models (NorESM-LM, NorESM-MM, CMCC-ESM2, and FIO-ESM-2-0) that replicate a strong correlation between the PQDO and AADO as Group 1, whereas another four models (NESM3, ACCESS-CM2, MPI-ESM1-2-LR, and MRI-ESM2-0) that indicate poor correlation as Group 2. The criterions selecting models are introduced in the Supplementary Table 2. The composited regression maps of the geopotential height and SAT onto the PQDO index in the Group 1 show a strong Rossby wave train propagating to the Ross-Amundsen Seas and the resultant surface warming (Fig. 5d). The suppressed ASL in response to the PQDO causes anomalous warm (cold) advection, reducing (increasing) SIC over the Ross-Amundsen Seas (Bellingshausen Sea) (Supplementary Fig. 8a). The simulated patterns of atmospheric circulation and surface warming in the Group 1 agree with those reproduced in the pacemaker experiment and the observation. The models in Group 2 simulate a rather weak tropical-polar

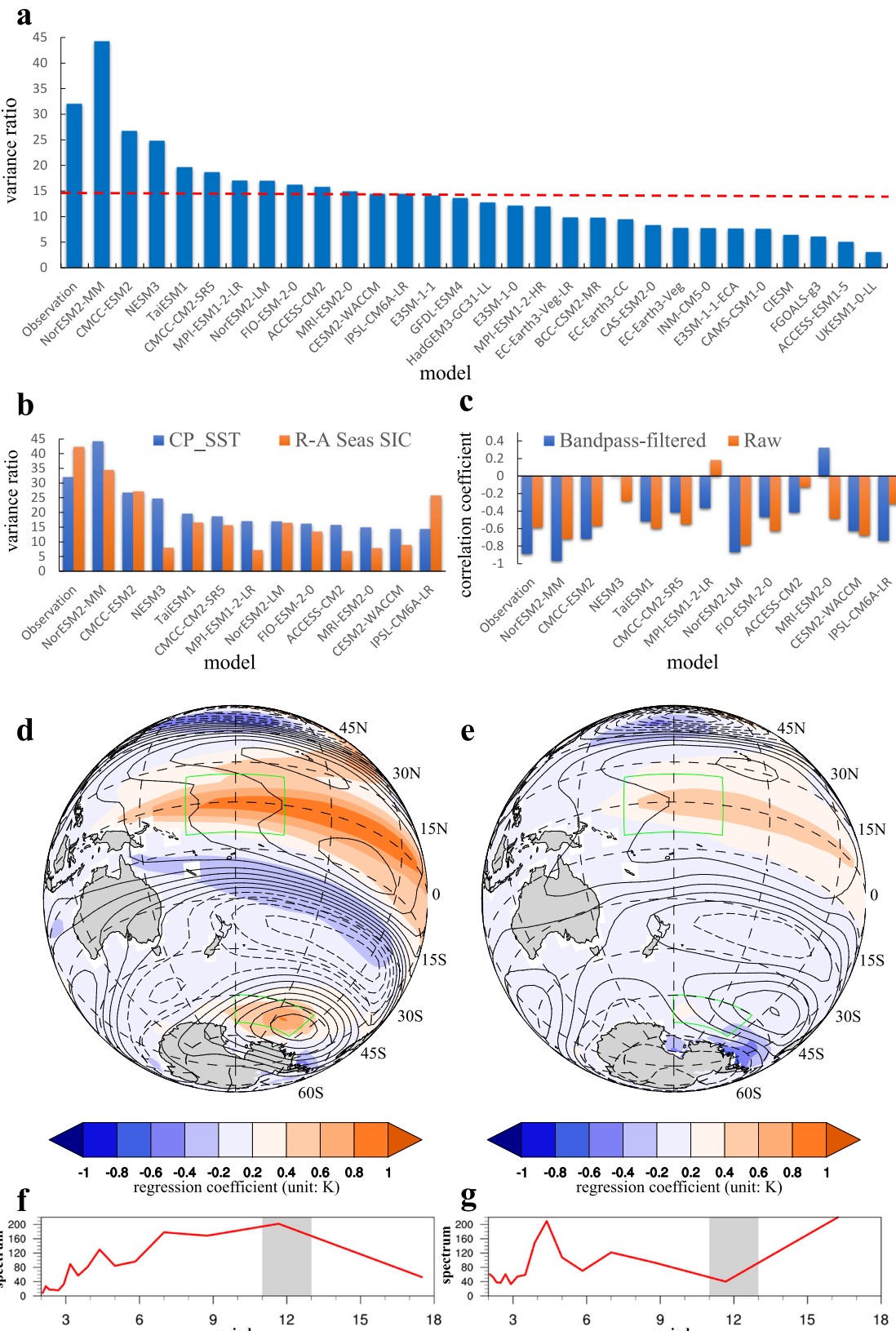

teleconnection and consequently show no significant warming over the Amundsen Sea (Fig. 5e). Over the Antarctic, the geopotential height pattern resembles the positive SAM, which leads to a circum-polar SIC reduction (Supplementary Fig. 8b) due to weakened westerlies and enhanced upwelling of warm deep water[3]. It is worth mentioning that the inability of models in Group 2 to simulate the wave trains could

reflect different mechanisms connecting the central tropical Pacific and the Antarctic, other than the stationary wave dynamics. As suggested in previous studies[22], the tropical Pacific SST may influence the SAM via atmospheric background circulation, such as the subtropical jet and the Hadley circulation, which may be reflected by the models in Group 2 to some extent. In addition, models that reproduce the Rossby

**Fig. 5 | Coupled Model Intercomparison Project historical simulations. a** The quasi-decadal variance ratio (%) of the central tropical Pacific sea surface temperature (SST) simulated in 29 Coupled Model Intercomparison Project Phase 6 models. The quasi-decadal variance ratio is defined as the variance of 8–16-yr bandpass filtered series divided by the variance of the raw series. **b** The quasi-decadal variance ratios (%) of the central tropical Pacific SST (CP_SST) and Ross-Amundsen Seas sea ice concentration (R-A Seas SIC) in the models that reproduce the Pacific quasi-decadal oscillation (PQDO). **c** The correlation coefficients between

the PQDO and the Antarctic decadal oscillation (AADO) in the models that reproduce the PQDO. **d, e** are the composited regression maps of surface air temperature (shading; unit: K) and 200 hPa geopotential height (contours; unit: m) onto the PQDO index in Group 1 and Group 2, respectively. The data has been preprocessed by 8–16-yr bandpass filtering. **f, g** are the composited global wavelet spectrums of the Ross-Amundsen Seas sea ice concentration in Group 1 and Group 2, respectively. Base maps for Fig. 5d, e were generated using the NCAR Command Language (version 6.6.2).

wave train associated with the PQDO could accurately replicate the quasi-decadal periodicities in Antarctic sea ice (Fig. 5f). The tropical periodic SST signal is sent to Antarctic sea ice by the poleward propagating wave train. The lack of the Rossby wave train in models of Group 2 suggests that the local circulation system—like the SAM—dominates the SIC variability, as there is no decadal periodicity (Fig. 5g). As a result, we can draw the conclusion that the quasi-periodic pattern of Antarctic sea ice variability is a result of a crucial poleward propagating Rossby wave train on the quasi-decadal timescale.

### Antarctic sea ice prediction model

This study demonstrates an oscillating signal in Antarctic sea ice originating from the central tropical Pacific SST and the associated tropical-polar teleconnection. Further, we employ the Monte Carlo Singular Spectrum Analysis to examine the credibility of the PQDO's quasi-periodicity. We generated 5000 realizations using an autoregressive model and the extended PQDO observation (1960–2020) since its quasi-periodicity only emerges till the late 1950s[45,48]. It exhibits a spectral peak within 12–14 years, which is statistically significant at the 95% confidence level (Supplementary Fig. 9). The result indicates that the PQDO is a robust mode of decadal oscillation. Its oscillating nature and projected fluctuations in Antarctic sea ice may provide a plausible approach to improve sea ice predictability on the decadal scale.

Here we construct a prediction model for the AADO:

$$AADO(t) = a \cdot PQDO(t) + c_1 \qquad (1)$$

$$PQDO(t) = b \cdot PQDO(t - \tau) + c_2 \qquad (2)$$

where $t$ denotes year. $a$ and $b$ are regression coefficients, and $c_1$ and $c_2$ are residual. The parameter $\tau$, estimated using the lagged autocorrelation of the PQDO index, indicates the time length of the PQDO phase transition. The oscillating nature of PQDO is shaped by the lagged influence of nonlinear dynamical heating related to ENSO nonlinearity[42], with a maximum lagged auto-correlation at approximately seven years ($\tau = 7$, Supplementary Fig. 10). Thus, knowing the current state, $PQDO(t - \tau)$, will inform us on its future changes (Eq. 2). Further, the AADO can be predicted seven years in advance based on its simultaneous relationship with the PQDO (Eq. 1).

Figure 6a, b exhibits two experimental cross-validated forecasts for the periods 2008–2015 and 2013–2020, respectively. The prediction model is trained by excluding the data from the forecasted periods. The results are preprocessed by an 8-yr low-pass filter to isolate the decadal variation. The sea ice prediction model successfully reproduces the maximum around 2010 and the drastic decline in the 2010s, showing considerable multi-year predicting skills. Additional six sets of forecasts, individually starting from 2009 to 2012, are conducted to evaluate the performance of this model (Supplementary Fig. 11). The correlation coefficients between predicted and observed series are ranging from 0.76 to 0.97, and over half of the forecasts pass the 90% confidence level. The capability to reproduce the multi-year sea ice variability is overall stable among forecasts, suggesting that the PQDO-based regression model is a robust approach to estimating the AADO seven years ahead, with considerable accuracy. In Fig. 6c, the

predicted AADO series suggests that the Ross-Amundsen SIC will recover in the next four to five years before it decreases once again since the mid-2020s.

## Discussion

In this study, we identify a decadal mode of Antarctic sea ice variability, referred to as the AADO. The AADO is modulated by the central tropical Pacific SST (the PQDO), exhibiting synchronized decadal fluctuations ($r = -0.90$) and a consistent spectral peak of 12–14 years. The PQDO excites a poleward propagating Rossby wave train, with a significantly weakened ASL at the downstream. The changes in local atmospheric circulation further embed the PQDO signal into the sea ice variation via thermodynamic feedbacks. The mechanism has also been consistently reproduced by the Central Pacific pacemaker experiment and CMIP6 historical simulations.

The initialization of sea ice conditions in the climate model is challenging due to a limited observational period and data quality. In this study, the PQDO-based statistical model can capture the multi-year variability of sea ice well. Here we highlight the importance of introducing the PQDO signal into climate model initializations when predicting Antarctic sea ice, which may be a plausible approach to further improve the multi-year predictability of sea ice in more sophisticated physical models. Moreover, among CMIP6 models, the performance in simulating the tropical-polar teleconnection influences the reproducibility of the Antarctic sea ice variations, which may have significant implications in reducing the model biases.

The PQDO-AADO relationship can be differentiated from the ENSO- and IPO-related tropical-polar teleconnections in terms of temporal behaviors. For ENSO, it variates primarily on interannual scales[55,56] (Supplementary Fig. 12a), whereas the PQDO shows significant decadal fluctuations (Supplementary Fig. 12b). The spectral coherence of the Ross-Amundsen Seas SIC with ENSO peaks within 2–6 years, whereas it is most significant in the frequency band of 8–16 years with the PQDO (Supplementary Fig. 12c, d), indicating a decadal connection that is significantly differentiated from ENSO. For the IPO, it shows a pronounced multidecadal variability (20–30 years)[57] (Supplementary Fig. 12e), but the periodic feature of IPO is obscure. Over 7% for the remaining SIC decadal anomaly in the Ross-Amundsen Seas can be explained by the PQDO after removing the IPO signal (Supplementary Fig. 12f), suggesting that the impacts of PQDO on the AADO are independent of the IPO. The conclusions drawn here do not necessarily contradict that ENSO and the IPO modulate Antarctic sea ice[18]. Still, they influence the Ross-Amundsen Seas sea ice differently from the PQDO primarily due to different temporal scales of variability. However, we do acknowledge a coherence between the PQDO and the North Pacific Gyre Oscillation (NPGO)[58], which shows considerable loading over the Northeast Pacific on decadal scales ($r = -0.69$), implying that Antarctic sea ice may be further influenced by the decadal signals from the extratropical North Pacific.

## Methods

### Data

In this study, the monthly SIC is derived for the period 1981–2020 from the National Snow and Ice Data Center (NSIDC) dataset. It covers the range from 45°S–90°, with a spatial resolution of 1° × 1°. The SIC is

# a. Prediction 2008-2015

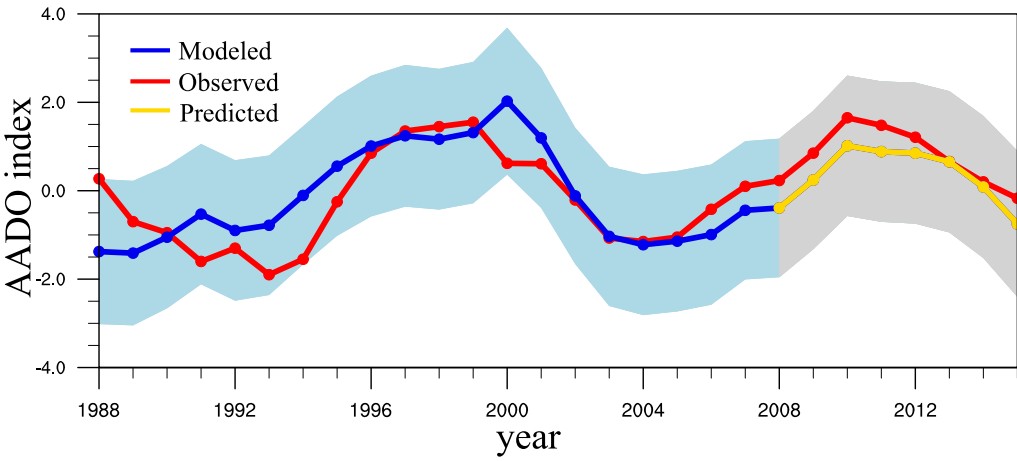

# b. Prediction 2013-2020

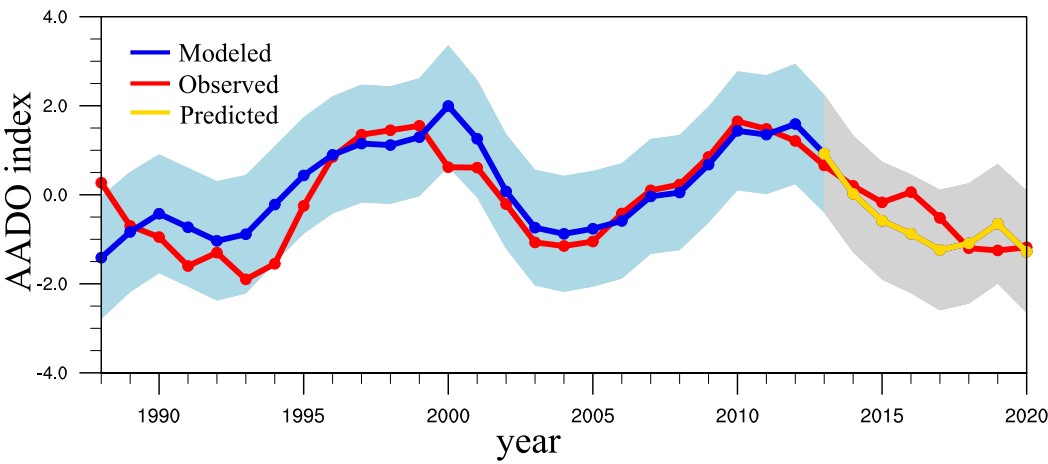

# c. Prediction 2020-2027

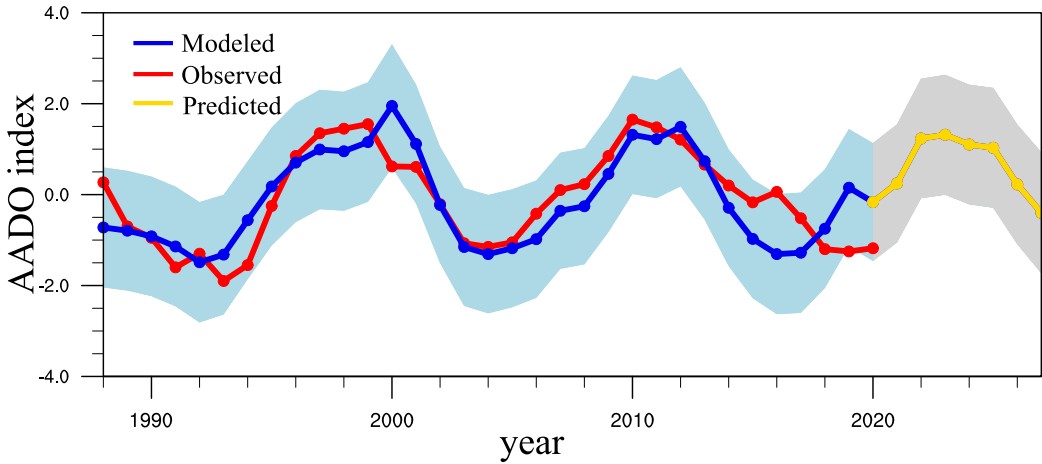

**Fig. 6 | Prediction model of the Antarctic decadal oscillation.** The observed (red), modeled (blue), and predicted (green) Antarctic decadal oscillation (AADO) index. **a** shows the hindcast from 2008 to 2015. **b** as in (**a**) but for the period 2013–2020. **c** is the predicted sea ice from 2020 to 2027 computed from the Pacific quasi-decadal oscillation-based prediction model. The data are preprocessed by an 8-yr lowpass filter to isolate the decadal variation. The shaded areas show the 2-sigma uncertainty range of the modeled and hindcasted/predicted values.

estimated from satellite-based observation of passive microwave brightness temperatures[59]. It is calibrated from different sensors and produces consistent long-term data to allow the analysis of sea ice variability. We also employ the OISST[60] and the HadISST[61] SIC datasets (both at a 1° resolution) in comparison with the NSIDC data. The sea ice motions are observed sea ice drift derived from the NSIDC Polar Pathfinder Daily 25 km EASE-Grid Sea Ice Motion Vectors, Version 4. The sea ice motion vectors are derived from satellite-based observation, buoy position data, and NCEP reanalysis data and have been widely used in previous studies[62,63]. This dataset has a spatial resolution of 25 km and covers from 37°S to 90°S. The data is available for the period 1978–2021. The SST data used to define the PQDO index is obtained from the Extended Reconstruction SSTs version 5 (ERSSTv5)[64], with a spatial resolution of 2° × 2°. The atmospheric data (surface air temperature, winds, and geopotential height) is derived from the ERA5 reanalysis[65] (regrided to the T63 grid in this study). Note that the radiative fluxes from the ERA5 reanalysis shows a large bias in the Southern Ocean[66], while the one from the NCEP2 Reanalysis (used in this study) has better performances[67] (defined as positive downward)[68]. The subsurface temperature is used for the period 1981–2020 derived from the EN4 reanalysis[69], which has a spatial resolution of 1° × 1° and 42 layers. The historical simulations from 29 CMIP6 models (listed in Supplementary Table. 1) are analyzed for the period 1981–2014 and regrided to the T42 grid.

### Index definitions
The AADO index is defined as the area-weighted average of SIC over the Ross-Amundsen Seas (60°S–75°S, 180°–125°W).

The PQDO index is defined as the area-weighted average of sea surface temperature in the central tropical Pacific (10°S–10°N, 165°E–165°W). This definition is consistent with previous studies[45,52].

The IPO index is defined as the PC2 of 8-yr lowpass filtered Pacific SST (120°E–120°W, 40°S–60°N), which is similar to the definition in the previous study[18].

The Nino 3 index is defined as the area-averaged SST anomalies (ERSSTv5) over the central-eastern tropical Pacific (5°N–5°S, 150°W–90°W). It is downloaded from https://psl.noaa.gov/enso/dashboard.html.

The NPGO index is defined as the second dominant mode of sea surface height variability in the Northeast Pacific, which can be downloaded from http://o3d.org/npgo/.

### Statistical methods
This study focuses on quasi-decadal variability, so the annual mean of monthly data is calculated before analysis. To identify the quasi-periodicities, we apply the local wavelet spectrum method[70,71] with a Morlet wavelet base, which has been widely employed in previous studies[52]. The cross-wavelet spectrum is also used so that we can quantify the coherence of periodicities as well as phases between time series[72]. The significance of the power spectrum is also examined using the Monte Carlo Singular Spectrum Analysis[73]. Here we use an autoregressive model to fit the observed data and generate 5000 realizations with the same parameters to show if the spectral peak is a real containing signal within a limited sample size.

We examine the separation of EOFs, following the "rule of thumb" in ref. 54, which is estimated using the following equations:

$$\lambda_1 - \lambda_2 \geq \delta\lambda_1 + \delta\lambda_2 \approx (\lambda_1 + \lambda_2) \cdot \sqrt{\frac{2}{N}} \tag{3}$$

where $\lambda_1$ and $\lambda_2$ are the eigenvalues, $\delta\lambda_1$ and $\delta\lambda_2$ are the corresponding sampling errors, and N is the sample size. It has been demonstrated by North et al. that the estimated sampling error is partly determined by sample sizes so that the test result is plausible with a large number of realizations.

In this study, we employ an 8-yr lowpass filter to isolate the decadal SIC variation while removing the interannual signal, such as ENSO. The statistical significance of the correlation coefficient is evaluated by a two-tail Student's t-test, where the effective number of degrees of freedom ($N^{eff}$) is estimated by:

$$\frac{1}{N^{eff}} \approx \frac{1}{N} + \frac{2}{N} \sum_{j=1}^{N} \frac{N-j}{N} \rho_{XX}(j)\rho_{YY}(j) \tag{4}$$

where N denotes the sample size and $\rho_{XX}(j)$ stands for the autocorrelation of the sampled time series X and $\rho_{YY}(j)$ is for Y, while j is the time lag.

### Sea ice advection
The SIC advection term is calculated using the following equation[74]:

$$SIC_{adv} = - \left( \mathbf{u} \cdot \frac{\partial SIC}{\partial x} + \mathbf{v} \cdot \frac{\partial SIC}{\partial y} \right) \tag{5}$$

where $\mathbf{u}$ and $\mathbf{v}$ are sea ice motions, x and y denote longitudes and latitudes. The advection term is calculated based on annual mean SIC and sea ice drift data.

### Pacemaker experiment
Here we use an intermediate complexity atmospheric general circulation model from the International Centre for Theoretical Physics (ICTPAGCM version 41, also called "SPEEDY")[75,76]. The AGCM is coupled to the Nucleus for European Modeling of the Ocean (NEMO) version 3 model[77], which includes a sea ice component (LIM version 3)[78]. The ICTPAGCM has eight vertical levels and a horizontal resolution of T30 (3.75° × 3.75°), with simplified parameterization schemes. The model is computationally efficient, while it exhibits on-par performance in simulating large-scale features and climate variability compared to state-of-the-art models[76]. The NEMO model solves primitive equations (z-coordinate) on a tripolar ORCA2 grid (horizontal resolution of 2° × 2°, and 0.5° × 0.5° in the tropics), and its physical ocean component contains both dynamics and thermodynamics. Considering the tight interaction between sea ice and the underlying ocean, the ocean dynamics is interfaced with the sea ice component that takes into account ice dynamics, thermodynamics, subgrid-scale thickness variations, and brine inclusions[79]. The ocean and sea ice components in the NEMO model are coupled to the ICTPAGCM via the OASISv3 coupler[80], allowing us to inspect the complex interactions within the ocean-atmosphere-sea ice system over the Antarctic Seas.

In the pacemaker experiment, the ICTPAGCM-NEMO coupled model is relaxed to the observed monthly varying SST (ERSSTv5) over the central tropical Pacific (10°S–10°N and 165°E–165°W), while allowing the model to freely evolve outside the central tropical Pacific region (referring to as CP_EXP). The pacemaker experiment aims to inspect the PQDO-related changes in atmospheric circulation and the associated ocean-atmosphere-sea ice coupling over the Antarctic. The SIC is modulated by dynamic and thermodynamic processes in the atmosphere as well as the underlying ocean. The model is conducted from 1955 to 2022 (1981–2020 for analysis). The experiment is started from an already spin-up state (for about 1000 years) with stabilized global mean surface air temperature, sea ice cover, and top-of-atmosphere net energy flux, while an additional 26-year is for the pacemaker experiment to spin up[81]. The carbon dioxide concentration is fixed in the CP_EXP so that we can isolate the internal forcing on the sea ice decadal variability.

## Data availability
All data are available publicly in the main text or the supplementary materials. NSIDC sea ice concentration data are available at https://nsidc.org/data/soac/sea-ice-concentration. NSIDC Polar Pathfinder

dataset is obtained from https://nsidc.org/data/nsidc-0116/versions/4. ERA5 Reanalysis is available at https://www.ecmwf.int/en/forecasts/dataset/ecmwf-reanalysis-v5. NCEP2 Reanalysis is available at https://psl.noaa.gov/data/gridded/data.ncep.reanalysis2.html. EN4 Reanalysis data is derived from https://www.metoffice.gov.uk/hadobs/en4/. CMIP6 historical simulations are downloaded from https://esgf-node.llnl.gov/projects/cmip6/.

## Code availability

The data and figures in this study were analyzed and produced with publicly available packages in NCAR Command Language (NCL), Python, and MATLAB. All base maps were generated using the NCL (version 6.6.2)[82]. The scripts are provided by the corresponding author upon requests.

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

## Acknowledgements

This research was jointly supported by the National Key Research and Development Projects (Grant No. 2022YFF0801703), the National

Natural Science Foundation of China (NSFC) Project (41975082,42130607), and the Fundamental Research Funds for the Central Universities (2233300001).

## Author contributions
Y.L. and C.S. designed the research. Y.L., C.S., F.K. and M.A.A. performed the data analysis, prepared all figures, and led the writing of the manuscript. Y.L., C.S., J.L., F.K., E.D.L., M.A.A. and X.L. discussed the results and commented on the manuscript.

## Competing interests
The authors declare no competing interests.
