## [Peer Review File · Nature Communications]

Decadal oscillation provides skillful multiyear predictions of Antarctic sea iceREVIEWER COMMENTS

Reviewer #1 (Remarks to the Author):

General comments:

This study examines decadal sea ice variability and predictability in the Ross and Amundsen Seas from the perspective of the atmospheric teleconnection associated with the decadal SST variability in the central tropical Pacific. The authors identify the Antarctic sea ice decadal oscillation (AADO) as the second EOF mode of the pan-Antarctic sea ice variability. However, the variance explained by the second EOF mode is small and close to the first EOF mode, and the spatial pattern of the second EOF mode appears to be out of phase by 60° compared to the first EOF mode. Given these aspects, the second EOF mode may not be well separated from the first EOF mode and not represent the pan-Antarctic decadal sea ice variability because it shows a large variability only in the Ross and Amundsen Seas.

The authors also conducted a coupled model pacemaker experiment and analyzed CMIP6 historical runs to claim that the Pacific Quasi-Decadal Oscillation (PQDO) can generate the Amundsen Sea Low variability through atmospheric teleconnection and hence sea ice variability in the Ross and Amundsen Seas. However, the spatial pattern of the atmosphere and sea ice variability is almost the same as in previous studies on the influence of the Interdecadal Pacific Oscillation (IPO) on the decadal sea ice increasing trend in the Pacific sector (e.g., Meehl et al. 2016). The results shown here are not novel as if the authors repeat the previous studies by introducing different climate indices. Considering the lack of new findings, I could not recommend this paper for a possible publication in this high-rank journal. Below are major concerns and specific comments that can be addressed to further improve this study.

1. Motivation of this study (L60-78)

Several studies already discussed a role of the Interdecadal Pacific Oscillation (IPO) in the Antarctic sea ice variability in the Pacific sector through atmospheric teleconnection (e.g., Meehl et al. 2016). Despite this growing evidence, why did the authors focus on the PQDO and revisit the same issue, although the conclusion is almost the same with other existing research? What is the difference with the impacts of the IPO?

- Meehl, G. A., Arblaster, J. M., Bitz, C. M., Chung, C. T., & Teng, H. (2016). Antarctic sea-ice expansion between 2000 and 2014 driven by tropical Pacific decadal climate variability. *Nature Geoscience*, 9(8), 590-595.

2. EOF analysis of Antarctic sea ice (L86-108)

Why did the authors apply the EOF to the annual mean SIC, not to the detrended SIC anomalies? PC1 (Supplementary Fig. 1b) shows an increasing trend, but it also exhibits a decadal variability (e.g., low sea ice in the 1980s, high sea ice in the early 1990s). The variance explained by PC1 (16.9%) is close to that by PC2 (13.1%), and the spatial pattern of EOF1 (Supplementary Fig. 1a) appears to be out of phase by 60° compared to the EOF2 (Fig. 1a). I am wondering if EOF1 and 2 are not well separated by the North criterion (North et al. 1982). What about the variance explained by PC3? Is PC2 well separated from PC3 as well?

- North, G. R., Bell, T. L., Cahalan, R. F., & Moeng, F. J. (1982). Sampling errors in the estimation of empirical orthogonal functions. *Monthly weather review*, 110(7), 699-706.

3. Correlation analysis of climate variables (L109-132)

The authors attributed the Amundsen Sea Low variability to the surface air temperature and sea ice variability in the Ross and Amundsen Seas through the correlation analysis, but the correlation does not explain causality between the two variables. For example, the higher SAT can generate the lower SIC through surface melt, while the lower SIC can induce the higher SAT through decrease in surface albedo. To demonstrate the causality, I would recommend the authors should perform lead-lag correlation analysis for the variables.

Also, it is unclear which of the shortwave radiation and warm air advection is more important for the sea ice decrease in the Ross and Amundsen Seas. Also, the authors examined sea ice drift only, but did not examine the sea ice melt from below due to subsurface ocean warming (e.g., Meehl et al. 2019). The authors should elaborate the physical processes on the sea ice variability in more quantitative ways (e.g., lead-lag regression analysis, sea ice volume tendency).

- Meehl, G. A., Arblaster, J. M., Chung, C. T., Holland, M. M., DuVivier, A., Thompson, L., ... & Bitz, C. M. (2019). Sustained ocean changes contributed to sudden Antarctic sea ice retreat in late 2016. *Nature Communications*, 10(1), 14.

4. Possible links with the PQDO (L134-L160)

A spatial pattern of SST variance on a quasi-decadal timescale (Fig. 3a) resembles the IPO related pattern. Also, the atmospheric teleconnection associated with the PQDO is almost the same as that with the IPO (e.g., Meehl et al. 2016). The difference in the physical processes between the PQDO and IPO is unclear from the sentence. It is hard to believe that the influence of the PQDO on the Amundsen Sea Low and sea ice variability is novel and different from that of the IPO.

5. Pacemaker experiment results (L161-175)

The pacemaker experiment captures the negative correlation between the PQDO and Ross sea ice variability (Fig. 4d), but does not simulate the negative correlation between the PQDO and Amundsen sea ice variability. This is inconsistent with the observational results (Fig. 4c), although the spatial pattern of the correlation with the geopotential height at 200 hPa (Fig. 4b) is similar to the observation (Fig. 4a). Does it mean that the central Pacific variability cannot explain the sea ice variability in the Amundsen Sea and there are other mechanisms (e.g., subsurface ocean warming) that control the sea ice variability?

6. CMIP6 models results (L176-210)

The selection of the models for the two groups is subjective, lacking the objective criterion. For example, TaiESM2 and CMCC-CM2-SR5 show the fourth and fifth largest variance ratios and negatively high correlations between the PQDO and AAO, but these models are not taken into account for Group 1. I am not sure why the authors selected FIO-ESM2.0 for Group 1, although this model does not show as high variance ratio as these two models.

7. Antarctic sea ice prediction model (L212-244)

The hindcast results from the model look overfitting to the observation (Fig. 6). The authors employed the auto-regression models of PQDO to predict the AADO, but some of the AADO signal may persist over years by internal dynamics independent of the PQDO. The authors can develop the auto-regression models of the AADO similarly as in Eq. (2), then check whether the prediction skills based on the PQDO are higher than the persistence prediction by the AADO itself. I am also wondering why the authors only show two experimental forecasts (2008-2015 and 2013-2020) rather than evaluating the prediction skills for all the forecasts starting from different initial years. This would help examine the robustness of the prediction accuracy based on the model.

8. Methodology on the pacemaker experiment (L310-336)

Before conducting the pacemaker experiment, does this coupled model really capture the Amundsen Sea Low and Antarctic sea ice variability when the model is forced by the global SST observation? If so, how much can the atmospheric and sea ice variability in the above control experiment be explained by the pacemaker experiment forced by the observed SST in the central tropical Pacific? It looks to me that the climatological SIC in the CP_EXP is not well simulated with a high sea ice bias (Supplementary Fig. 11).

Specific comments:

L32: hereafter "central tropical Pacific".

L54: Are these "intermediate" climate models?

L57: Remove "and Amundsen".

L104-105: The authors should rethink the name of AADO, because the sea ice variability is pronounced mostly in the Ross and Amundsen Seas. Can it be rephrased as "Ross and Amundsen decadal variability"?

L124: sea ice drift.

L124: How did the authors calculate the sea ice advection term? Is it from the sea ice reanalysis or the pacemaker experiment?

L147-148: Correlation analysis cannot explain causality between the two variables.

L164-166: This statement is incorrect. The sea ice only in the Ross Sea declines.

L178: It appears to be 9 out of 29 modes from the red line in Fig. 5a.

L179-180: How did the authors select the six or nine models from Figs. 5b and c?

L187: other four models.

L202-202: The meaning of the sentence is unclear.

L216: Why did the authors use the extended PQDO observation back to 1960, although there are no sea ice data before 1980s?

L234: respectively (Figs. 6a, b).

L244: the mid-2020s (Fig. 6c).

L313: Which version of NEMO and LIM did the authors use?

L316: on-par performance?

L324: in the Antarctic Seas.

L330-332: The argument is incorrect. The CP_EXP shows higher SIC climatology than the observation.

L333-334: 21 years are not enough for spinning up the model in the extratropical regions.

Reviewer #2 (Remarks to the Author):

Liu et al. describe a mode of quasi-decadal Antarctic sea ice variability defined as the second component of the empirical orthogonal function of the observed Antarctic sea ice concentration over the satellite era. The mode explains relatively well the variability of Antarctic sea ice while being mostly significant in the northeastern Ross Sea and the Amundsen Sea. The authors link the so-called Antarctic sea ice Decadal Oscillation (AADO) to the Pacific quasi Decadal Oscillation (PQDO), a regional mode of SST variability in the central tropical Pacific, via the generation and propagation of atmospheric Rossby waves modulating the strength of the Amundsen Sea Low and impacting the state of the sea ice in this region. The authors make use of a variety of datasets: remotely-sensed sea ice concentration and SST datasets for the identification of leading modes of Antarctic sea ice and Pacific

SST variability, atmospheric reanalysis to identify drivers of sea ice variability, a selection of CMIP6 historical simulations and a dedicated pacemaker experiment. This abundance and diversity of datasets contribute to increasing the confidence in the authors' findings and help build physical understanding.

To my knowledge, it is the first time that this link between the PQDO and Antarctic sea ice is identified and described. Antarctic sea ice variability is still poorly understood, and tropical-polar teleconnections are. The present work thus represents a significant advance in the understanding of Antarctic sea ice variability and its tropical teleconnections, potentially explaining recent observations and serving as a basis for projections. The authors indeed propose a simple forecast model that seems able to accurately reproduce the evolution of the AADO index.

General comments:

While the definition of the AADO mode, and the relationships presented in this study seem robust and convincing, the authors only discuss the linkage with the PQDO, and not with other modes SST variability in the tropical Pacific. Indeed, similar weakening of the ASL via Rossby wave trains on interannual to decadal timescales imputable for instance to ENSO have already been described (Li et al. 2021). The separation between the PQDO and ENSO is quickly discussed in the paper, but the implications of the proximity between ENSO and the PQDO in terms of results interpretation and how the analysis strategy is preventing misinterpretation should be made more clear in the manuscript.

Finally, the paper is overall well written and presented, but there is sometimes a lack of precision or inadequate wording. The paper would therefore benefit from careful reading and editing. The figures are well chosen and properly contribute to the argumentation, but their overall quality can be improved (size of labels and legends, missing units, map orientation, caption ...).

Detailed comments

L22: "Over the satellite era 1981-2020, Antarctic sea ice exhibits a substantial increasing trend [...]". I wouldn't consider the Antarctic sea ice extent trend as "substantial", see for instance Yuan et al. (2017). In any case, it contradicts the statement made at line 24: "[...] the dramatic decline of Antarctic sea ice in 2014-2018 [...]"

L28 (and elsewhere in the manuscript): "anticorrelated" would be more precise than "out of phase".

L54: "intermediate climate models" → unclear

L60: "The low-frequency oscillatory phenomena have [...]" → "Low-frequency oscillatory phenomena have [...]"

L62: "interdecadal~multidecadal" → "inter- to pluri-decadal"

L64: "One of which" → "One of these decadal modes"

L65: "warming and cooling phases over time" → "warming and cooling phases".

L66: "referring" → "referred"

L72: "fundamentally different from the dynamic nature of ENSO" → "fundamentally different from ENSO".

L80: "The underlying mechanism is inspected based on the pacemaker experiment using the ICTPAGCM-NEMO coupled model implemented with a dynamic sea ice module and historical simulations from CMIP6" → "The underlying mechanism is inspected in historical CMIP6 simulations and a dedicated pacemaker experiment conducted with the ICTPAGCM-NEMO coupled model".

L87: what is the dataset used here?

L89: "The corresponding PC1 shows an overall icing trend" → "an overall increase in sea ice cover". Can the positive trend of PC1 be directly translated into an increase in sea ice cover, since both positive and negative correlations with PC1 exist?

L91: "The EOF2" → "The second mode of EOF"

L96: "12~14" → "12-14"

L100: "The decline of SIC from 2012 and relict from 2018 may be a part of the current cycle" → Precise what do you mean by "SIC" here: is it the Antarctic sea ice extent? sea ice concentration in the Ross Sea? The AADO index?

L103: "Antarctic sea ice" → "Antarctic sea ice variability".

L105: it would be valuable to state here how you define the AADO index (as stated in the "Index definitions" part.

L109: "thermodynamic/dynamics" → "thermodynamical and dynamical"

L109: "associated with the changes in large-scale atmospheric circulations" → "largely driven by large-scale atmospheric circulation"?

L112: "AADO strongly coincides with local surface air temperature (SAT)" → as SAT is also strongly responding to SIC changes, SAT warming could be explained by a decrease in SIC induced by other factors. SAT warming still coincides with warm advection and SW radiations though.

L124: what do you mean by "sea ice draft effect"?

L124: "The advection term" → Advection of what? "Sea ice advection"?

L134: "Antarctic sea ice is closely" → "Antarctic sea ice variability is closely"

L150: "convections" → "convection"

L167: "the center of sea ice" → unclear

L177: how do you define the "quasi-decadal variance ratio"

L179: "There are six of them" → "Six of them "

L182: "correlation coefficients greater than 0.4" → "negative correlation coefficients lower than -0.4".

L183: "those models" → "the models"

L183: "above average" → the multi-model mean has little value here, I would rather say that the models with the more realistic intensity of SST variance and Antarctic Ross sea ice quasi-decadal variability simulate significant correlations between the PQDO and AADO index.

L186: "coherence" → correlation?

L186: "Group. 1" → "Group 1"

L188: "Group. 2" → "Group 2"

L190: the pattern in Figure 5d looks a lot like ENSO. In general, how can you confidently differentiate the PQDO from other modes of variability, such as ENSO or the IPO?

L195: "For the models in Group 2, they" → "The models in Group 2 simulate"

L201: "wave trains, on the other hand, reflects" → "wave trains could reflect"

L203: "imitate" → "reproduce" ?

L206: remove "on the other hand"

L206: "The lack of the Rossby wave train," → "The lack of the Rossby wave train in models of group 2"

L221: "scale" → "timescale"

L230: "lead us to" → "inform us on"?

L241: "In recent years" → which years?

The references to sea ice increase or decrease in this paragraph are vague and somehow misleading: are you referring to the sea ice concentration in the Ross-Amundsen sea (i.e., your AADO index), or the Antarctic sea ice extent?

L242: "relict tendency" → "reverse tendency"?

L242: "modulated by"

L247: rephrase.

L250: "downstream portion, which significantly weakens" → downstream portion significantly weakening the Amundsen Sea Low.

L253: "suggested" → "simulated" or "reproduced"

L259: "the sea ice concentration will recover" → where? Precise that this is suggested by the PQDO-based prediction model.

L275: remove "used in this study"

L276: "using algorithms" → this is very vague, either be more precise or remove this sentence while keeping the reference.

L278: "Other sea ice data are employed from" → We also employ the OISST and the HadISST sea ice concentration dataset (both at a 1° resolution) "

L279: "which are overall consistent" → What do you mean by "consistent" here?
L280: "is from" → "is obtained from"
L283: why do you use fluxes from NCEP2 and atmospheric states from ERA5? Wouldn't it be more consistent to use a single dataset here?
L313: LIM represents both dynamical and thermodynamical sea ice processes. What is the version of LIM used in your model?
L317: "compared to state-of-the-art models" → reference needed.
L326: "tropical central pacific" → where exactly?
L339: unless I have missed something, the model data of the pacemaker experiment does not seem to be made available. The authors could indicate where to find the other data used in the study (Observations, reanalyses, CMIP6 simulations ...)

Figures:

Figure 1c: labels are tough to read.
Figure 2a: Caption: "The correlation maps of (a) the AADO index (multiplied by -1), (b) domain-averaged surface air temperature (SAT) with 200 hPa geopotential height" → "The correlation maps of (a) the AADO index (multiplied by -1) and (b) surface air temperature (SAT) averaged over the Ross Sea (green box) with 200 hPa geopotential height"
Figure 3a: missing unit/label for the SST variance color bar.
Figure 3b: labels are tough to read, missing x-axis labels or units, and y-axis units.
Figure 4: Missing labels on the color bar
Figure 4b: Keep consistency between the legend (AADO) and figure caption (Ross-Amundsen Sea SIC)
Figure 4c and d: using the same color bar scale would ease comparison.
Figure 6: what is shown on the vertical axis? the caption says "antarctic sea ice", which is unclear, are you referring to the AADO index? Antarctic sea ice extent?
Figures with maps: (Figures 1a, Fig sup. 1a, 2a, 2b, 4a, 4b, 8a, 8b, 11a, and 11b) I would recommend using the standard map orientation (lon=0) on top to make the map easier to read.

References:

Yuan et al. (2017), Increase of the Antarctic Sea Ice Extent is highly significant only in the Ross Sea, Scientific Reports.

Reviewer #3 (Remarks to the Author):

Title: Decadal oscillation provides skillful multiyear predictions of Antarctic sea ice
Authors: Liu et al.

Overview:

This manuscript focuses on the mechanisms and predictability of Antarctic sea ice on decadal timescales. The authors analyzed observational datasets and coupled climate model experiments to identify a strong link between the low-frequency variability of Antarctic sea ice and tropical Pacific variability on quasi-decadal timescales. The manuscript also presents a simple statistical model that demonstrates the predictability of this variability several years in advance.

I believe that the manuscript's findings are highly significant and valuable in advancing our understanding of Antarctic sea ice variability and improving climate predictability on longer timescales. Given the importance of Antarctic sea ice as a metric for climate change assessment and the challenges of simulating it accurately in global climate models, these insights are particularly valuable.

The manuscript is clearly and logically presented, with strong writing throughout. While I have a few

suggestions that could help readers better understand the content, these do not affect the manuscript's conclusions. Therefore, I recommend a minor revision. Please find my detailed comments below.

Minor comments:

1. Line 67: "ENSO": It appears to be the first use of the acronym, and therefore it should be spelled out.
2. Line 124: "The advection term": Please clarify which advection it is. I believe it would be "the sea-ice advection," but I am unsure whether it is the vertical, zonal, or meridional advection due to the mean flow or anomaly flow.
3. Line 161: "In the pacemaker experiment": This term appears suddenly. It would be better to refer to the method section to avoid confusing the reader.
4. Line 209: Remove "probably."
5. Discussion section: Please include a few sentences or a paragraph to discuss climate prediction using a dynamical model. The authors analyzed fully coupled climate models (the pacemaker experiment and CMIP6 models), but the prediction model is a simple statistical model. Initializing the sea-ice condition in the climate model is challenging due to the observational data period and quality. However, according to the results in this manuscript, the sea-ice condition in the climate model may be initialized using the tropical Pacific SST only, and such a system may demonstrate multi-year predictability. The impact of model performance on predictability could also be discussed from the perspective of CMIP6 simulations. Such discussion would enhance the manuscript's importance to a broader community.
6. Lines 317-319: Please include the horizontal resolution of the NEMO model.
7. Lines 325-326: "the observed SST over the tropical Central Pacific": Please include the latitude-longitude domain for the tropical Central Pacific. The data source for the observed SST is also required. The method is also unclear: is it nudged to the observed SST, used the heat flux, or replaced with the observed SST?

The response to the reviewers of the manuscript
“Decadal oscillation provides skillful multiyear predictions
of Antarctic sea ice”

Revised to *Nature Communications*

August 2023

Reviewer #1:

General comments:

This study examines decadal sea ice variability and predictability in the Ross and Amundsen Seas from the perspective of the atmospheric teleconnection associated with the decadal SST variability in the central tropical Pacific. The authors identify the Antarctic sea ice decadal oscillation (AADO) as the second EOF mode of the pan-Antarctic sea ice variability. However, the variance explained by the second EOF mode is small and close to the first EOF mode, and the spatial pattern of the second EOF mode appears to be out of phase by 60° compared to the first EOF mode. Given these aspects, the second EOF mode may not be well separated from the first EOF mode and not represent the pan-Antarctic decadal sea ice variability because it shows a large variability only in the Ross and Amundsen Seas.

The authors also conducted a coupled model pacemaker experiment and analyzed CMIP6 historical runs to claim that the Pacific Quasi-Decadal Oscillation (PQDO) can generate the Amundsen Sea Low variability through atmospheric teleconnection and hence sea ice variability in the Ross and Amundsen Seas. However, the spatial pattern of the atmosphere and sea ice variability is almost the same as in previous studies on the influence of the Interdecadal Pacific Oscillation (IPO) on the decadal sea ice increasing trend in the Pacific sector (e.g., Meehl et al. 2016). The results shown here are not novel as if the authors repeat the previous studies by introducing different climate indices. Considering the lack of new findings, I could not recommend this paper for a possible publication in this high-rank journal. Below are major concerns and specific comments that can be addressed to further improve this study.

Reply: We would like to express our sincere gratitude towards the reviewer's constructive comments and suggestions, which greatly improve the quality of our manuscript. We consider the issues outlined in the comments carefully and have done our best to address the reviewer's concerns. We provide more observational and modeling evidence that would significantly enhance the rigorousness of our analysis.

The results demonstrate that the decadal oscillation is a robust mode of the Ross-Amundsen Seas sea ice concentration (SIC), and its teleconnection with the Pacific quasi-decadal oscillation (PQDO) can be differentiated from existing studies. Point-by-point responses to the reviewer's comments (in italics) are listed below.

1. Motivation of this study (L60-78)

Several studies already discussed a role of the Interdecadal Pacific Oscillation (IPO) in the Antarctic sea ice variability in the Pacific sector through atmospheric teleconnection (e.g., Meehl et al. 2016). Despite this growing evidence, why did the authors focus on the PQDO and revisit the same issue, although the conclusion is almost the same with other existing research? What is the difference with the impacts of the IPO?

- Meehl, G. A., Arblaster, J. M., Bitz, C. M., Chung, C. T., & Teng, H. (2016). Antarctic sea-ice expansion between 2000 and 2014 driven by tropical Pacific decadal climate variability. Nature Geoscience, 9(8), 590-595.

Reply: We appreciate the reviewer's comments. The teleconnections between the tropical Pacific and Antarctic climate via the stationary Rossby wave have been widely recognized ¹ on the interannual ^{2,3} and multidecadal scales ^{4,5}. However, the quasi-decadal variability of Antarctic sea ice and its relation to the tropical SSTs are still not fully understood. In this study, we find, for the first time, a quasi-periodic variation in the Ross-Amundsen Seas sea ice concentration (SIC), referring to as the AADO, which cannot be explained by either El Niño Southern Oscillation (ENSO) ⁶ or the Interdecadal Pacific Oscillation (IPO) ⁵. The Pacific quasi-decadal oscillation (PQDO) ⁷ modulates the Ross-Amundsen Seas sea ice variation, exhibiting synchronized decadal fluctuations as in SIC, which may be independent of existing tropical-polar teleconnections.

In Meehl et al. 2016 ⁵, the IPO index is defined as the PC2 of 13-yr lowpass filtered Pacific SST. However, the AADO shows a spectral peak of around ten years. Therefore, the decadal variance in sea ice cannot be fully captured by the IPO index used in previous studies ⁸⁻¹⁰. Here we employ the same EOF method but with 8-yr lowpass

filtered Pacific SST. The SST data has been detrended so that the PC1 is referred to as the IPO index. Considering that the quasi-periodicity of PQDO has only emerged since the 1950s^{7,11}, the IPO index is calculated from 1950 to 2020. Note that the IPO index used here is not sensitive to the defining periods and shows consistent temporal features in the overlapping period as that defined since 1900⁵.

The difference between the IPO and the PQDO

The predominant variance of the IPO and the PQDO varies on temporal scales. In Fig. A1, the IPO shows a pronounced multidecadal variability (20–30 years)¹², which is characterized by an increasing trend from the 1950s to the 1980s and an opposite trend afterward. The IPO has two phase shifts around 1970 and 2000 that are consistently shown in Meehl et al. 2016⁵. The periodic feature of IPO is obscure for the analyzed period. While the IPO captures the multidecadal variability, the PQDO exhibits more significant quasi-decadal fluctuations, with a spectral peak at 8–16 years. The PQDO has experienced over four quasi-cycles since the late 1950s, which is significantly different from the IPO series. The correlation coefficient between the PQDO and the IPO is also small ($r = 0.23$), demonstrating that the PQDO and the IPO may not be a same thing.

Fig. A1 Normalized time series of the PQDO index (blue line) and the IPO index (black line) for the period 1950–2020. The trend has been removed from the PQDO index.

Furthermore, the spatial patterns of the IPO and the PQDO are separable. In Fig. A2a,

the regressed pattern of IPO shows considerable loading over the Kuroshio-Oyashio Extension (KOE) region and the eastern tropical Pacific. The positive IPO corresponds with SST warming over the central-eastern Pacific and an arch-shaped cooling over the subtropical northwestern and South Pacific. Fig. A2b is calculated by regressing the residual of the IPO-related pattern onto the PQDO index so that the IPO signal is statistically removed here. The PQDO exhibits prominent loading over the central tropical Pacific, Northeast Pacific, and subtropical Pacific, which can be separated from the IPO pattern to some extent. In addition, the SST signal in response to the PQDO is stronger than to the IPO, especially over the central tropical Pacific, American west coast, and Gulf of Alaska, which resembles the quasi-decadal variance (8–16 years) pattern shown in Fig. 3a (in the manuscript). It suggests that the quasi-decadal signal in the Pacific could largely be explained by the PQDO rather than the IPO since the IPO variates primarily on multidecadal timescales.

Fig. A2 (a) The regression of 8-yr lowpass filtered SST onto the IPO index for the period 1950–2020 (unit: K). The residual is computed by subtracting the IPO-regressed component from the Pacific SST; (b) The regression of the residual onto the 8-yr lowpass filtered PQDO index (unit: K).

Mechanisms behind the PQDO are also different from that of the IPO. Previous studies regarded the IPO variability as an integration of the stochastic ENSO activity^{13,14}, while Meehl and Hu 2006 suggested that the wind-driven ocean Rossby waves and subtropical cells are responsible for the IPO¹⁵. For the PQDO, Liu et al. 2022 proposed that nonlinear dynamical heating energized by super El Niño events shapes the quasi-

decadal periodicity of the PQDO ¹⁶., while the two-way coupling between the central tropical Pacific and extratropical North Pacific may also play an important role ^{17,18}. Investigating the mechanism that underlies the PQDO may be beyond the scope of the current study. Nevertheless, the predominant PQDO variance lies in the timescale between 8 and 16 years, which is different from the interannual ENSO and the multidecadal IPO signals. More importantly, the PQDO displays a significant periodic fluctuation that is synchronized to the Ross-Amundsen Seas sea ice variability on the decadal scale.

Impacts on Antarctic sea ice

The AADO has shown approximately three cycles since the satellite observation, with alternative increasing and decreasing trends that last for about 4–8 years (Fig. A3). While the AADO exhibits significant quasi-periodic variability on the decadal scale, the primary temporal feature of IPO is a multidecadal oscillation that consists of an increasing trend from 1950 to 1980 and a decreasing trend from 1980 to 2020 (Fig. A1). The correlation coefficient between the AADO and the IPO for the overlapping period only reaches -0.18. The phase of IPO shifted from positive to negative around 2000, which coincides with the decline of AADO from 1998 to 2003. However, in the mid-2000s, the AADO reclaims against the continuous decline of the IPO, indicating that the phases between them are not well matched due to different scales of variability. Thus, the quasi-periodicity of the Ross-Amundsen Seas sea ice could not be regulated solely by the IPO.

The poleward propagating Rossby wave is an essential pathway that transmits tropical signals toward the Antarctic ¹. It has been observed on a variety of temporal scales, including the interannual ENSO ^{2,3} and the multidecadal IPO ^{4,5}. At the downstream of the wave train, atmospheric circulation anomalies (e.g., the Amundsen Sea Low, ASL) modulate sea ice via local dynamic and thermodynamic processes. Here we show that the intensity of ASL also exhibits quasi-periodic fluctuations over the last 40 years, which corresponds well with the sea ice variability. As shown in Supplementary Fig. 5 (in the manuscript), the ASL index has a predominant variance on the decadal scale rather than a multidecadal shift found in the IPO index. The correlation coefficient

between the IPO and the ASL indices is 0.20, indicating that the IPO-related atmospheric teleconnection cannot explain the quasi-periodicity in the ASL. A weak relationship can also be found between the IPO and SAT ($r = 0.27$) over the Ross-Amundsen Seas. Thus, the local sea ice-atmosphere feedbacks responsible for the AADO are unlikely driven by the IPO either.

Fig. A3 The 8-yr lowpass filtered and detrended series of the PQDO (blue line), AADO (red line), ASL (green line), and SAT (gold line; area-averaged SAT over the Ross-Amundsen Seas) indices for the period 1981–2020.

We further inspect the spatial coherence between Antarctic sea ice and the PQDO after removing the IPO signal. The IPO signal is removed by subtracting the IPO-regressed SIC from the raw data. The residual is then regressed onto the PQDO index. As shown in Fig. A4, over 7% of the remaining sea ice decadal variability over the Ross-Amundsen Seas can be explained by the PQDO, and the overall pattern somewhat resembles the EOF2 of Antarctic sea ice (Fig. 1a in the manuscript), suggesting that the impacts of PQDO on Antarctic SIC are independent of the IPO. Based on the above analysis, we may conclude that the decadal oscillation of the Ross-Amundsen Seas sea ice is likely driven by the PQDO, which shares a consistent quasi-periodicity of 8–16 years. The PQDO-related SIC changes can be separated from that of the IPO temporally and spatially. However, the conclusions drawn here do not necessarily contradict

previous studies which suggested that the IPO modulates multidecadal trends in Antarctic sea ice⁵. The PQDO and the IPO influence the Ross-Amundsen Seas sea ice differently, primarily due to different temporal scales of variability.

Fig. A4 The regression of Antarctic SIC onto the 8-yr lowpass filtered PQDO index (unit: 1) after removing the IPO signal. The IPO signal is removed by subtracting the IPO-regressed SIC from the raw data. The residual is then regressed onto the PQDO index.

Moreover, the AADO-related SST pattern (Fig. A5a) overall agrees with that of the PQDO (Fig. A2b) but also is similar to the North Pacific Gyre Oscillation (NPGO)¹⁹ regime as there are also considerable loadings over the subtropical North Pacific and Northeast Pacific. The NPGO index is defined as the second EOF of North Pacific sea surface height (Fig. A5b). The correlation coefficient between the PQDO index and the NPGO index is -0.69, implying that the PQDO may be a part of the NPGO dynamics, which can also be statistically and physically separated from the IPO to some extent. We would like to discuss the possible link between the AADO and NPGO, which may have further implications for the Pacific-Antarctic teleconnections. (please see lines 297-351)

Fig. A5 (a) The correlation map between the 8-yr lowpass filtered SST and the AADO index for the period 1981–2020. Trends have been removed from the SST and the AADO index; (b) The normalized and detrended series of the PQDO index and the NPGO index for the period 1950–2020.

2. EOF analysis of Antarctic sea ice (L86-108)

Why did the authors apply the EOF to the annual mean SIC, not to the detrended SIC anomalies? PC1 (Supplementary Fig. 1b) shows an increasing trend, but it also exhibits a decadal variability (e.g., low sea ice in the 1980s, high sea ice in the early 1990s). The variance explained by PC1 (16.9%) is close to that by PC2 (13.1%), and the spatial pattern of EOF1 (Supplementary Fig. 1a) appears to be out of phase by 60° compared to the EOF2 (Fig. 1a). I am wondering if EOF1 and 2 are not well separated by the North criterion (North et al. 1982). What about the variance explained by PC3? Is PC2 well separated from PC3 as well?

- North, G. R., Bell, T. L., Cahalan, R. F., & Moeng, F. J. (1982). *Sampling errors in the estimation of empirical orthogonal functions*. *Monthly weather review*, 110(7), 699-706.

Reply: Thank you. In order to extract the key features of Antarctic sea ice (both trends and variations), we apply the EOF to the raw data. Following the reviewer’s suggestions, we further discuss the statistical independence of the EOFs.

Fig. A6 is the lead-lag correlation of PC1 and PC2. The simultaneous correlation coefficient is 0.09, and the lead-lag relationship is also poorly coherent, suggesting that the PC1 and PC2 can be separated. Furthermore, we calculate the spatial pattern correlation between the EOF1 and the EOF2, and the correlation coefficient is 0. The patterns and corresponding variations between EOF1 and EOF2 are not correlated, suggesting that they are spatially and temporally distinguishable to some extent.

Fig. A6 The lead-lag correlation between the PC1 and the PC2 of Antarctic SIC for the period 1981–2020.

We examine the separation of EOFs, following the “rule of thumb” in North et al ²⁰, which is estimated using the following equations:

$$\lambda_1 - \lambda_2 \geq \delta\lambda_1 + \delta\lambda_2 \approx (\lambda_1 + \lambda_2) \cdot (2/N)^{1/2}$$

where λ_1 and λ_2 are the eigenvalues, $\delta\lambda_1$ and $\delta\lambda_2$ are the corresponding sampling errors, and N is the sample size. It has been demonstrated by North et al. that the estimated sampling error is partly determined by sample sizes so that the test result is plausible with a large number of realizations. In this study, only 39 years are available using annual SIC data, which may lead to an overestimation of sampling error. Thus,

we apply the EOF on monthly SIC for the period 1981-2020, with an extended 468-realization. The EOFs can be well separated via the North criterion, as the EOF1 and EOF2 explain approximately 9% and 6% SIC variance, respectively. In Fig. A7, the patterns of the monthly EOFs are in good agreement with those of the annual EOFs. The pattern correlation coefficient between the annual and monthly EOF1 patterns is 0.90 (Fig. A7a and A7c), with considerable loadings over the Weddell Sea, eastern Ross Sea, and the Amundsen Sea. For the monthly EOF2 pattern (Fig. A7d), the primary loading is located over the Ross-Amundsen Seas, which is consistently shown in the annual EOF2 (Fig. A7b; pattern correlation = 0.74). Thus, we may conclude that the spatial patterns between EOF1 and EOF2 are statistically separated, and the spatial characteristics are different.

Fig. A7 (a) EOF1 and (b) EOF2 patterns for the annual Antarctic SIC as in the manuscript; (c) and (d) as in (a) and (b), but for the monthly Antarctic SIC.

In Supplementary Fig. 1, the primary feature of PC1 is a long-term increasing trend superimposed on interannual variability (2–8 years). The shift from the 1980s to the 1990s may subsume into a long-term increasing trend rather than a decadal fluctuation. The phases between PC1 and PC2 are not well corresponded. To separate the trend and

variability, we further inspect the EOF with detrended annual SIC as suggested by the reviewer. After removing the trend, PC1 mainly denotes the low-frequency variations of Antarctic sea ice (Fig. A8a), explaining about 19% variance, and strongly correlates with the original PC2 using raw data ($r = 0.77$). In Fig. A8b, the PC1 shows significant spectral peaks at 8–16 years, revealing a decadal oscillation as that shown in the manuscript. The low-frequency component of PC1 is extracted via an 8-yr lowpass filtering (Fig. A8c). It has experienced approximately three cycles from 1981 to 2020, which are opposite to the PQDO index ($r = -0.70$).

Based on the above analysis, we may conclude that the decadal oscillation is one of the leading modes of Antarctic sea ice variability (most pronounced in the Ross-Amundsen Seas), which can be separated, spatially and temporally, from other variations. (please see lines 99-101 and 103-109)

Fig. A8 (a) The PC1 of the detrended annual Antarctic SIC (red line) and the PC2 of the raw annual Antarctic SIC (blue line); (b) The power spectrum of the PC1 of the detrended annual Antarctic SIC; (c) The 8-yr lowpass filtered PC1 of the detrended annual Antarctic SIC (red line) and the PQDO index (blue line).

3. Correlation analysis of climate variables (L109-132)

The authors attributed the Amundsen Sea Low variability to the surface air temperature and sea ice variability in the Ross and Amundsen Seas through the correlation analysis, but the correlation does not explain causality between the two variables. For example, the higher SAT can generate the lower SIC through surface melt, while the lower SIC can induce the higher SAT through decrease in surface albedo. To demonstrate the causality, I would recommend the authors should perform lead-lag correlation analysis for the variables.

Also, it is unclear which of the shortwave radiation and warm air advection is more important for the sea ice decrease in the Ross and Amundsen Seas. Also, the authors examined sea ice drift only, but did not examine the sea ice melt from below due to subsurface ocean warming (e.g., Meehl et al. 2019). The authors should elaborate the physical processes on the sea ice variability in more quantitative ways (e.g., lead-lag regression analysis, sea ice volume tendency).

*- Meehl, G. A., Arblaster, J. M., Chung, C. T., Holland, M. M., DuVivier, A., Thompson, L., ... & Bitz, C. M. (2019). Sustained ocean changes contributed to sudden Antarctic sea ice retreat in late 2016. *Nature Communications*, 10(1), 14.*

Reply: Thank you for your great comments. We inspect subsurface temperature over the Ross-Amundsen Seas, as pointed out by the reviewer that it may influence the sea ice. The area-averaged Ross-Amundsen Sea subsurface temperature profile is calculated for the period 1981–2020 using the EN4 reanalysis ²¹.

As shown in Fig. A9a, the subsurface sea temperature exhibits a significant interannual variability, accompanied by the vertical movements of warm water that are affected by surface wind stress curl and Ekman suction. It has been demonstrated that the upward shift of subsurface warm water from 2005 to 2020 may reflect its variabilities on decadal/multidecadal scales ²². The ups and downs of warm water correspond with warmer and cooler near-surface ocean temperature so that we can examine its temporal evolution with sea temperature at a depth of 25m (Fig. A9b). The low-frequency components of the near-surface ocean temperature and the AADO are related to some

extent ($r = -0.52$) since the subsurface warming would cause sea ice to melt. However, the near-surface ocean temperature exhibits significant inter- to multi-decadal fluctuations, with two warming trends in the 1980s and 2010s and a cooling trend during 1990-2010, rather than a quasi-periodic variation found in sea ice. The power spectrum of near-surface ocean temperature shows no peak at 8-16 years (Fig. A9c). Furthermore, we calculate the lead-lag correlation between the AADO index and near-surface ocean temperature (Fig. A9d) using monthly data. The simultaneous correlation coefficient reaches -0.22 , consistently showing the influence of subsurface warming on sea ice melting. However, the peak occurs when the near-surface ocean temperature lags sea ice by three months ($r = -0.26$), indicating that the feedback of sea ice melting to subsurface temperature through surface albedo may be nonnegligible. The subsurface temperature indeed contributes to the low-frequency variations of Antarctic sea ice melting, but the coherence between them may involve a two-way interaction, which is beyond the scope of the current study. Nevertheless, the decadal oscillation found in the Ross-Amundsen Seas sea ice cannot be fully explained by the subsurface temperature variability since its quasi-periodicity is not significant.

Fig. A9 **(a)** The vertical profile of detrended subsurface temperature averaged over the Ross-Amundsen Seas from 1981 to 2020; **(b)** The normalized and 8-yr lowpass filtered series of the AADO index and the near-surface temperature (at 25m) averaged over the Ross-Amundsen Seas; **(c)** The power spectrum of the near-surface temperature index; **(d)** The lead-lag correlation between the AADO index and the near-surface temperature index, which is calculated from the monthly data. The x-axis denotes the lead months of near-surface temperature.

In addition to subsurface temperature, we reexamine the sea ice horizontal advection using satellite-based sea ice drift observation from the NSIDC dataset (see specific comment L124). The sea ice advection shows a drastic long-term decreasing trend imposed on low-frequency variations from 1984 to 2016 (Fig. A10a). The correlation coefficient between the AADO index and area-averaged SIC advection only reaches 0.11. An overall in-phase relationship can be observed, but the opposite tendencies in the 2010s may weaken their coherence. Through lead-lag correlation (Fig. A10b), the relationship between SIC and SIC advection peaks when the advection leads by one month ($r = 0.23$), and the simultaneous correlation coefficient can also reach 0.21. Thus, the advection can be regarded as a forcing term, slightly contributing to the sea ice changes. However, the dynamic effect is unlikely to be a dominant factor of the AADO, considering the weak temporal coherence between their low-frequency components.

Fig. A10 (a) The normalized and 8-yr lowpass filtered time series of the Ross-Amundsen Seas SIC advection and the AADO index; (b) The lead-lag correlation between the AADO index and the SIC advection, which is calculated from the monthly data. The x-axis denotes the lead months of SIC advection.

As shown in Fig. 2c (in the manuscript), surface air temperature (SAT) strongly correlates with SIC ($r = -0.97$), exhibiting a synchronized quasi-periodic variation. Here we further calculate the lead-lag correlation to identify the causality between SAT and SIC. In Fig. A11a, the distribution of lead-lag correlation coefficients deviates to the left, with greater correlation coefficients when SAT leads than lags, SIC by 1 to 3 months. In other words, the SIC melting driven by preceding and contemporary SAT warming is stronger than its feedback to maintain the anomalous SAT. Although the lagged response of SAT may not be fully excluded here, the SAT warming, induced by the weakened ASL, still has an important triggering effect on sea ice melting. In the revised manuscript, we will discuss the interaction between SAT and SIC. (please see lines 145-154)

The SAT is governed by the ASL mainly via shortwave radiation and atmospheric temperature advection, which are strongly correlated with the AADO as well. By

constructing a multivariate regression model of the AADO, we can evaluate the relative contributions of shortwave radiation and temperature advection to the decadal variation of sea ice. The regression model is constructed as follows:

$$AADO(t) = a \cdot SW(t) + b \cdot T_{adv}(t) + c$$

where a , and b are coefficients, c is residual, and t is year. $AADO(t)$, $SW(t)$, and $T_{adv}(t)$ denote the Ross-Amundsen Seas SIC, shortwave radiation, and air temperature advection at 850 hPa, respectively. The regression model overall fits the observed AADO ($r = 0.88$), showing consistent decadal oscillation (Fig. A11b). The standardized partial regression coefficients for the shortwave radiation and temperature advection are -0.83 and -0.13 , suggesting that the radiative heating term plays a more important role in modulating SIC over the Ross-Amundsen Seas. (please see lines 125-129 and 135-139)

Fig. A11 (a) The lead-lag correlation between the AADO index and the SAT index, which is calculated from the monthly data. The x-axis denotes the lead months of SAT;

(b) The time series of the observed and reconstructed AADO index.

We've examined the temporal coherence of the AADO with subsurface temperature, sea ice advection, and SAT. Both subsurface temperature and sea ice advection may influence the low-frequency variability of Ross-Amundsen Seas sea ice. However, the quasi-periodicity found in the AADO is obscure in those variables, which are unlikely the driving factors of such a decadal oscillation. Meanwhile, SAT warming explains over 90% low-frequency variance of the Ross-Amundsen Seas SIC. It is likely a driving factor of the decadal oscillation in SIC, with synchronized quasi-periods of 8–16 years.

4. Possible links with the PQDO (L134-L160)

A spatial pattern of SST variance on a quasi-decadal timescale (Fig. 3a) resembles the IPO-related pattern. Also, the atmospheric teleconnection associated with the PQDO is almost the same as that with the IPO (e.g., Meehl et al. 2016). The difference in the physical processes between the PQDO and IPO is unclear from the sentence. It is hard to believe that the influence of the PQDO on the Amundsen Sea Low and sea ice variability is novel and different from that of the IPO.

Reply: Thank you for your comments. The relationship between the PQDO and the IPO is discussed in major comment 1.

5. Pacemaker experiment results (L161-175)

The pacemaker experiment captures the negative correlation between the PQDO and Ross sea ice variability (Fig. 4d), but does not simulate the negative correlation between the PQDO and Amundsen sea ice variability. This is inconsistent with the observational results (Fig. 4c), although the spatial pattern of the correlation with the geopotential height at 200 hPa (Fig. 4b) is similar to the observation (Fig. 4a). Does it mean that the central Pacific variability cannot explain the sea ice variability in the Amundsen Sea and there are other mechanisms (e.g., subsurface ocean warming) that control the sea ice variability?

Reply: Thank you for your comments. The simulated SIC shifts westward, showing a

significant negative correlation over the Ross Sea but a relatively weak and reversed response over the Amundsen Sea. Here we find that model biases in simulating the atmospheric circulation downstream of the tropical-polar Rossby wave train may cause the shifts of SIC.

The simulated patterns of atmospheric circulation and SAT are overall consistent with the observation (Fig. A12a and A12b). The SAT warming (cooling) over the Ross-Amundsen Seas and the Indian Ocean (the Weddell-Bellingshausen Seas) corresponds with the anomalous high (low). However, the primary bias is that the simulated high over the Amundsen Sea (indicated by the green box) deviates to the west compared with the observed one. The weakened ASL is a dominant local factor influencing SAT and SIC. The westward shift of high anomalies in the experiment consequently leads to the deviations of shortwave radiation (Fig. A12c) and temperature advection (Fig. A12d). The enhanced radiative heating and warm air advection are most significant over the Ross Sea while showing relatively weak responses over the Amundsen Sea, resulting in the bias of local SIC.

Nevertheless, the model successfully reproduces the local atmosphere-sea ice interaction. The weakened ASL causes sea ice to melt via shortwave radiation and temperature advection. More importantly, the model captures the decadal oscillation in both SIC and SAT, showing a significant quasi-periodicity synchronized to the PQDO (Fig. 12e). The correlation coefficient between SIC and SAT is -0.96, which is close to the observed relationship. Therefore, the bias in the Amundsen Sea SIC originates from the bias in simulating the local anomalous high, while the associated mechanisms overall agreed with the observation. In the revised manuscript, we will discuss the model bias in atmospheric circulation and its impacts on the simulated sea ice responses. (please see lines 193-198)

Fig A12 The correlation maps of the PQDO index with the SAT (shading) and geopotential height (contours) at 200hPa in the (a) CP_EXP and (b) observation, respectively; The correlation maps of the PQDO index with the (c) shortwave radiation and (d) temperature advection at 850hPa (8–16-yr bandpass filtered) in the CP_EXP; (e) The normalized and 8-yr lowpass filtered series of the simulated AADO index and the SAT averaged over the Ross-Amundsen Seas.

6. CMIP6 models results (L176-210)

The selection of the models for the two groups is subjective, lacking the objective

criterion. For example, TaiESM2 and CMCC-CM2-SR5 show the fourth and fifth largest variance ratios and negatively high correlations between the PQDO and AAO, but these models are not taken into account for Group 1. I am not sure why the authors selected FIO-ESM2.0 for Group 1, although this model does not show as high variance ratio as these two models.

Reply: Thank you for your comments. In this section, we tend to inspect whether the AADO and its teleconnection to the PQDO can be captured by CMIP6 models. The premise is that models can reproduce the PQDO to some extent, with the quasi-decadal variance ratio of the central tropical Pacific SST greater than 14%. Then, the selection of models for Group 1 and Group 2 synthetically considers the quasi-decadal variance ratio of SIC and the correlation between PQDO and AADO. Scoring metrics based on the correlation coefficients of PQDO-AADO are employed for models that are capable (or incapable) of reproducing the variance ratio in SIC.

First, we need to compare the reproducibility of the relationship between PQDO and AADO among models. The scoring metric is defined as follows:

$$score(i) = 0.7 * [-R_{raw}(i)] + 0.3 * [-R_{bandpass}(i)]$$

where the score of the model (i) is a weighted combination of the correlation coefficients between PQDO and AADO using raw and 8–16-yr bandpass filtered data. Despite that the filtered data highlights the decadal components, the raw series contains more signal that is important to comprehensively evaluate their coherence. The scores are listed from largest to smallest in Table. A1. For models with lower scores, the simulated relationships between PQDO and AADO are relatively weak, which are sorted into Group 2. On the other hand, the NorESM2-LM, NorESM2-MM, CESM2-WACCM, CMCC-ESM2, and FIO-ESM-2-0 are the top five models with higher scores, indicating that the PQDO-AADO teleconnection is relatively well reproduced. Additionally, models with high scores should also be capable of capturing the low-frequency signals in the Ross-Amundsen Seas SIC, which requires a decadal variance ratio greater than 10%. Thus, only NorESM2-LM, NorESM2-MM, CMCC-ESM2, and FIO-ESM-2-0 are selected for Group 1. (please see the note near Supplementary Table 2)

Table. A1

Model	Variance ratio of the AADO	Score	Note
NorESM2-LM	16.38	0.814	Group 1
NorESM2-MM	34.34	0.795	Group 1
Observation	42.22	0.68	
CESM2-WACCM	8.87	0.665	
CMCC-ESM2	27.05	0.615	Group 1
FIO-ESM-2-0	13.42	0.582	Group 1
TaiESM1	16.49	0.576	
CMCC-CM2-SR5	15.57	0.511	
IPSL-CM6A-LR	25.72	0.446	
MRI-ESM2-0	7.8	0.247	Group 2
ACCESS-CM2	6.75	0.217	Group 2
NESM3	7.92	0.206	Group 2
MPI-ESM1-2-LR	7.15	-0.015	Group 2

7. Antarctic sea ice prediction model (L212-244)

The hindcast results from the model look overfitting to the observation (Fig. 6). The authors employed the auto-regression models of PQDO to predict the AADO, but some of the AADO signal may persist over years by internal dynamics independent of the PQDO. The authors can develop the auto-regression models of the AADO similarly as in Eq. (2), then check whether the prediction skills based on the PQDO are higher than the persistence prediction by the AADO itself. I am also wondering why the authors only show two experimental forecasts (2008-2015 and 2013-2020) rather than evaluating the prediction skills for all the forecasts starting from different initial years. This would help examine the robustness of the prediction accuracy based on the model.

Reply: Thank you. Following the reviewer's comments, we conduct six sets of PQDO-based experimental forecasts, starting from 2008 to 2013, individually. The forecast

starts from 2008 to ensure enough samples for model training. In addition, the AADO shows a shift from an upward trend to a continuous decline around 2010. Thus, it is important to examine whether this prediction model is able to capture such a drastic change.

The correlation coefficients between the estimated and observed AADO for each predicting period are provided to evaluate the performance of this model (Fig. A13). The correlation coefficients range from 0.76 to 0.97, and over half of the forecasts pass the 90% confidence level. The capability to reproduce the multi-year sea ice variability is overall stable among forecasts, suggesting that the PQDO-based regression model is a robust approach to estimating the sea ice for years ahead, with considerable accuracy. (please see lines 270-276)

Fig. A13 The correlation coefficients between the observed and predicted AADO index during the six predicting periods.

We also examined the performance of the auto-regression model of the AADO, which is constructed as follows:

$$AADO(t) = \mathbf{a} \cdot AADO(t - \boldsymbol{\tau}) + \mathbf{b}$$

where **a** and **b** are coefficient and residual, respectively, and t denotes year. The parameter $\boldsymbol{\tau}$ is set to 7 years in the auto-regression mode of the AADO so that we can compare it with the PQDO-based model on the same predicting scale. Considering the relatively short observational periods of sea ice, the forecasts are limited. To make sure

there are enough samples to estimate the coefficients, we conduct two experimental forecasts for the periods 2012–2019 and 2013–2020. The corresponding correlation coefficients between the estimated and observed SIC are 0.87 and 0.63, respectively. The auto-regression model of AADO that relies on its multi-year persistence can reproduce the observed SIC variability to some extent, but it is less skillful than the PQDO-based model. It further suggests that the prediction skill of the AADO originates from the oscillating signal of the PQDO on a scale of 7 years.

In this section, we intend to discuss the implications of the PQDO-AADO teleconnection and their oscillatory features to improve the decadal predictability of Antarctic sea ice. As suggested in previous studies²³, the decadal predictability of the Ross-Amundsen Seas sea ice remains relatively low, even with refined model initializations of local SST, SIC, subsurface temperature, and salinity. The initialization of sea ice conditions in the climate model is also challenging due to a limited observational period and data quality. In this study, we find, for the first time, a robust mode of decadal oscillation in the Ross-Amundsen Seas SIC, which is driven by the central tropical Pacific SST. The relationship between them and the oscillatory features can be reproduced by some CMIP6 models. The PQDO-based simple statistical model can capture the multi-year variability of sea ice well. We highlight the importance of introducing the PQDO signal into climate model initializations when predicting Antarctic sea ice, which may have significant implications to further improve the multi-year predictability of sea ice in more sophisticated physical models.

8. Methodology on the pacemaker experiment (L310-336)

Before conducting the pacemaker experiment, does this coupled model really capture the Amundsen Sea Low and Antarctic sea ice variability when the model is forced by the global SST observation? If so, how much can the atmospheric and sea ice variability in the above control experiment be explained by the pacemaker experiment forced by the observed SST in the central tropical Pacific? It looks to me that the climatological SIC in the CP_EXP is not well simulated with a high sea ice bias (Supplementary Fig. 11).

Reply: We appreciate the reviewer's helpful suggestions. It is of importance to examine whether the coupled model is able to capture the observed atmosphere-sea ice interaction over the Antarctic through a global SST-forced control experiment. However, conducting this experiment using a fully coupled model costs a lot of computational resources and could be inefficient, considering the limited time we have. Despite that, we conduct an additional experiment where the model is prescribed with a significantly extended range within 45°S–45°N and 30°E–70°W, which covers the whole Indo-Pacific SST region at lower latitudes (Indo-Pacific SST-forced experiment). The experiment is conducted from 1955 to 2022. Note that the coupled model is started from an already spin-up state (for about 1000 years) with stabilized global mean surface air temperature, sea ice cover, and top-of-atmosphere net energy flux, while an additional 26-year is for the pacemaker experiment to spin-up. Considering that the decadal variance (8–16 years) of the Atlantic SST is comparably weak (Fig. 3a in the manuscript), its influence on the model performance in simulating the decadal oscillation in Antarctic sea ice may be negligible. Here we mainly focus on the Indo-Pacific region, which contains primary decadal signals of the global SST in the observation.

The correlation map between the PQDO index and SAT/geopotential height shows a consistent pattern as that in the observation. In Fig. A14a, heating in the central tropical Pacific excites a poleward propagating Rossby wave train. The centers of geopotential height correspond well with the observation, especially the weakened Amundsen Sea Low downstream of this wave train, suggesting that the coupled model is able to reproduce the observed tropical-polar teleconnection. The SAT warming signal is located east of the weakened ASL, indicating that the local mechanism related to temperature advection and shortwave radiation plays a role. This experiment also successfully reproduces the reversed relationship between the sea ice and the tropical SST forcing. The SIC over the Ross Sea and the Amundsen Sea declines significantly in response to the PQDO in the Indo-Pacific experiment, and the spatial patterns are overall agreed with the observation, despite that its intensity is slightly underestimated (Fig. A14b).

More importantly, the Indo-Pacific SST-forced experiment captures the low-frequency variability in the atmosphere-sea ice coupled system over the Ross-Amundsen Seas. As shown in Fig. A14c, the series of Ross-Amundsen Seas SIC exhibits a significant decadal oscillation, with a spectral peak within 8-16 years (Fig. A14d). The simulated Ross-Amundsen Seas SIC has consistent phases and peaks as in the observation. The SIC is well correlated with the ASL and local SAT ($r = -0.75$ and -0.97 , respectively), which exhibit a synchronized quasi-periodicity (Fig. A14e). The above results suggest that the coupled model has the capability to simulate the decadal oscillation in Antarctic sea ice and the associated local variables, as well as the tropical-polar teleconnection.

Fig. A14 (a) The correlation map of the PQDO index with the SAT (shading) and geopotential height (contours) at 200hPa in the Indo-Pacific SST-forced experiment; (b) The correlation between the PQDO index and the SIC over the Ross-Amundsen Seas; (c) The normalized series of the AADO index, the ASL index, and the SAT averaged over the Ross-Amundsen Seas. All data used here is processed by an 8-yr lowpass filter; The power spectrums of (e) SAT and (d) SIC over the Ross-Amundsen Seas, respectively.

Further, we can compare the results between the CP_EXP and Indo-Pacific experiments. Despite the significant differences that can be found in the locations of ASL, the SIC averaged over the Ross-Amundsen Seas shows consistent decadal fluctuations. The

Ross-Amundsen Seas SIC simulated in CP_EXP explained over 60% quasi-decadal (8-16-yr) variance of that in the Indo-Pacific experiment, further suggesting that the central tropical Pacific SST is the primary source of decadal variability of the Ross-Amundsen Seas SIC. We must note that the model biases can be reduced with more observational signals being introduced into the experiment. Thus, it is reasonable that the SIC pattern in the Indo-Pacific experiment seems closer to the observed pattern. For example, the climatology of Antarctic SIC in the Indo-Pacific experiment shows slightly reduced high biases compared to the CP_EXP (Fig. 15), despite that biases still exist compared to the observation. (please see lines 187-190 and the note near Supplementary Fig. 6)

Fig. 15 The climatology of Antarctic SIC (unit: %) in (a) the Indo-Pacific SST-forced experiment and (b) the observation, respectively. The climatology is computed for the period 1981–2020.

In conclusion, the ICTPAGCM-NEMO coupled model experiment is able to capture the decadal variability in the Ross-Amundsen Seas SIC and the associated SAT and geopotential height, further highlighting the important role the central tropical Pacific SST played in its formation. Nevertheless, disagreements between model simulation and observation can be found, which may be related to internal model biases and uncertainties. In the revised manuscript, we will also discuss the possible sources of model biases.

Specific comments:

L32: hereafter “central tropical Pacific”.

Reply: Thank you. It has been revised throughout the manuscript.

L54: Are these “intermediate” climate models?

Reply: Thank you. In reference 31, the model they used to evaluate the predictability of sea ice is LOVECLIM1.2, which is an intermediate complexity Earth system model. In reference 32, the output from CMIP5 models was analyzed. In the revised manuscript, we’ll make the statement clearer and recheck the citation here.

L57: Remove “and Amundsen”.

Reply: Thank you. We’ve made changes to this point.

L104-105: The authors should rethink the name of AADO, because the sea ice variability is pronounced mostly in the Ross and Amundsen Seas. Can it be rephrased as “Ross and Amundsen decadal variability”?

Reply: Thank you. We agree with the reviewer that the decadal variability is most significant in the Ross-Amundsen Sea, where we define the AADO index to capture the strongest oscillatory signal. Furthermore, the decadal variability of SIC is also the leading mode of pan-Antarctic SIC after detrending, which explains 19% SIC variance, suggesting that the decadal oscillatory feature is a robust mode of variability. The name of AADO is referred to as a decadal oscillation in the Antarctic, which is most pronounced in the Ross-Amundsen Seas, but it also has implications for the decadal oscillatory features elsewhere in the Antarctic Seas, such as the Bellingshausen Sea and the Indian Ocean.

L124: sea ice drift.

Reply: Thank you. It has been revised.

L124: How did the authors calculate the sea ice advection term? Is it from the sea ice reanalysis or the pacemaker experiment?

Reply: Thank you for your comments. The SIC advection term is calculated using the following equation ²⁴:

$$SIC_{adv} = -(u \cdot \frac{\partial SIC}{\partial x} + v \cdot \frac{\partial SIC}{\partial y})$$

where u and v are sea ice motions, x and y denote longitudes and latitudes. The advection term is calculated based on annual mean SIC and sea ice drift data. Note that the sea ice motions in the original manuscript are estimated using surface winds, which may introduce a large bias. In the revised manuscript, the sea ice motions are observed sea ice drift derived from the NSIDC Polar Pathfinder Daily 25km EASE-Grid Sea Ice Motion Vectors, Version 4. The sea ice motion vectors are derived from satellite-based observation, buoy position data, and NCEP reanalysis data and have been widely used in previous studies ^{25,26}. The spatial correlation is used to estimate the sea ice motion between image pairs acquired from the satellites. This dataset has a spatial resolution of 25km and covers from 37°S to 90°S. The data is available for the period 1978–2021. The description of this dataset is added to the revised manuscript and the discussion about sea ice drift can be found in major comment 3.

L147-148: Correlation analysis cannot explain causality between the two variables.

Reply: Thank you. The causal relationships of SIC with atmospheric and oceanic variables are discussed in major comment 3 and have been added to the revised manuscript (please see lines 125-129).

L164-166: This statement is incorrect. The sea ice only in the Ross Sea declines.

Reply: Thank you for pointing this out. The statement has been revised. The possible causes of the biases in simulating SIC responses over the Amundsen Sea are discussed in major comment 5.

L178: It appears to be 9 out of 29 modes from the red line in Fig. 5a.

Reply: Thank you. The red line, which denotes the 14% variance ratio of the central tropical Pacific SST, was incorrectly marked in the figure. We've made changes to this figure.

L179-180: How did the authors select the six or nine models from Figs. 5b and c?

Reply: Thank you. The criteria selecting models for Group 1 and Group 2 are discussed in major comment 6. We've made it clearer in the manuscript.

L187: other four models.

Reply: Thank you. It has been revised.

L202-202: The meaning of the sentence is unclear.

Reply: Thank you for your comment. For the models in Group 2, the simulated tropical-polar teleconnection is comparably weak, and the pattern of local atmospheric circulation resembles the positive Southern Annular mode (SAM), which is not well agreed with the observation and the results from Group 1. However, this disagreement may reveal a different mechanism connecting the central tropical Pacific and the Antarctic, other than the stationary wave dynamics. As suggested in previous studies ¹, the tropical Pacific SST may influence the SAM via atmospheric background circulation, such as the subtropical jet and the Hadley circulation, which may be reflected by the models in Group 2 to some extent.

L216: Why did the authors use the extended PQDO observation back to 1960, although there are no sea ice data before 1980s?

Reply: Thank you. Based on the analysis in the manuscript, the quasi-periodicity of the Ross-Amundsen Seas SIC is likely driven by the PQDO through the Rossby wave train. Considering the limited observational data of SIC, we use the PQDO index from 1960 to ensure there are as many samples as possible so that the test on the power spectrum is robust and significant. Another reason that we extend the data back to 1960 is that the quasi-periodicity of the PQDO only emerges till the late 1950s, as suggested by the

previous study ^{7,11}.

L234: respectively (Figs. 6a, b).

Reply: Thank you. It has been revised.

L244: the mid-2020s (Fig. 6c).

Reply: Thank you. It has been revised.

L313: Which version of NEMO and LIM did the authors use?

Reply: Thank you. The NEMO version 3 and the LIM3 sea ice model are employed in this study.

L316: on-par performance?

Reply: Thank you. The reference is added here ²⁷.

L324: in the Antarctic Seas.

Reply: Thank you. It has been revised.

L330-332: The argument is incorrect. The CP_EXP shows higher SIC climatology than the observation.

Reply: Thank you. The statement has been revised here. The bias in simulating the SIC climatology is discussed in the revised manuscript. In the CP_EXP, the model is only driven by the central tropical Pacific SST, which may cause bias. In the Indo-Pacific experiment (see comment 8), such a bias in simulating SIC climatology is reduced compared to the CP_EXP.

L333-334: 21 years are not enough for spinning up the model in the extratropical regions.

Reply: Thank you. The experiment is conducted from 1955 to 2021 (1981-2020 for analysis). The experiment is started from an already spin-up state (for about 1000 years)

with stabilized global mean surface air temperature, sea ice cover, and top-of-atmosphere net energy flux, while an additional 26-year is for the pacemaker experiment to spin-up. The experiment configuration is close to that in Kucharski et al. 2016²⁸. In the revised manuscript, we will make a clearer statement. (please see lines 407-411)

Reviewer #2 (Remarks to the Author):

Liu et al. describe a mode of quasi-decadal Antarctic sea ice variability defined as the second component of the empirical orthogonal function of the observed Antarctic sea ice concentration over the satellite era. The mode explains relatively well the variability of Antarctic sea ice while being mostly significant in the northeastern Ross Sea and the Amundsen Sea. The authors link the so-called Antarctic sea ice Decadal Oscillation (AADO) to the Pacific quasi Decadal Oscillation (PQDO), a regional mode of SST variability in the central tropical Pacific, via the generation and propagation of atmospheric Rossby waves modulating the strength of the Amundsen Sea Low and impacting the state of the sea ice in this region. The authors make use of a variety of datasets: remotely-sensed sea ice concentration and SST datasets for the identification of leading modes of Antarctic sea ice and Pacific SST variability, atmospheric reanalysis to identify drivers of sea ice variability, a selection of CMIP6 historical simulations and a dedicated pacemaker experiment. This abundance and diversity of datasets contribute to increasing the confidence in the authors' findings and help build physical understanding.

To my knowledge, it is the first time that this link between the PQDO and Antarctic sea ice is identified and described. Antarctic sea ice variability is still poorly understood, and tropical-polar teleconnections are The present work thus represents a significant advance in the understanding of Antarctic sea ice variability and its tropical teleconnections, potentially explaining recent observations and serving as a basis for projections. The authors indeed propose a simple forecast model that seems able to accurately reproduce the evolution of the AADO index.

Reply: We really appreciate the reviewer's support and accurate summary of this work. The reviewer's comments are very detailed, constructive, and insightful, which help us greatly improve the quality of our manuscript. Following the reviewer's suggestions, we add more discussion about the relationship between the PQDO-AADO

teleconnection and ENSO. The manuscript and figures are carefully revised according to the reviewer's comments, and please see the point-to-point responses below.

General comments:

1. While the definition of the AADO mode, and the relationships presented in this study seem robust and convincing, the authors only discuss the linkage with the PQDO, and not with other modes SST variability in the tropical Pacific. Indeed, similar weakening of the ASL via Rossby wave trains on interannual to decadal timescales imputable for instance to ENSO have already been described (Li et al. 2021). The separation between the PQDO and ENSO is quickly discussed in the paper, but the implications of the proximity between ENSO and the PQDO in terms of results interpretation and how the analysis strategy is preventing misinterpretation should be made more clear in the manuscript.

Reply: Thank you for the reviewer's helpful comments. Following the reviewer's comments, we compare the key characteristics between the Pacific quasi-decadal oscillation (PQDO) and El Niño Southern Oscillation (ENSO) and the corresponding teleconnection patterns toward the Antarctic. Here we employ the Niño 3 index, which is defined as the area-averaged sea surface temperature (SST) over 5°N–5°S and 150°W–90°W using the ERSSTv5 dataset. Our results suggest that the PQDO and its teleconnection with the Antarctic decadal oscillation (AADO) relationship can be separated from the ENSO signal due to different temporal scales of variability.

The temporal scales of variability and underlying mechanisms between ENSO and the PQDO are distinguishable. We first inspect the power spectrums of the Niño 3 and the PQDO indices. ENSO, indicated by the central-eastern tropical Pacific SST, varies primarily on interannual scales, with spectral peaks over 2-8 years (Fig. B1a)^{29,30}, whereas the PQDO shows a prominent decadal variation, with a spectral peak at 12–14 years (Fig. B1b)⁷. Accordingly, the mechanisms modulating the oscillatory features of ENSO and the PQDO may be different. Several mechanisms have been proposed to explain the 2–6-yr quasi-periodicity of ENSO, such as the recharge-discharge

mechanism^{31,32} and the delayed oscillator³³. For the PQDO, Liu et al. 2022 proposed that nonlinear dynamical heating energized by super El Niño events shapes the 8–16-yr quasi-period of PQDO¹⁶, while the two-way coupling between the central tropical Pacific and extratropical North Pacific also plays an important role^{17,18}. The above analysis indicates that the primary difference between the PQDO and ENSO depends on the temporal scales.

We then examine the power spectrum of the AADO index (Fig. B1c). The SIC over the Ross-Amundsen Seas has a spectral peak at 12–14 years, which corresponds well with that of the PQDO index, showing a synchronized decadal oscillation. Meanwhile, the Ross-Amundsen Seas SIC also exhibits periodicities of 2–6 years, which may be modulated by the interannual signals, such as ENSO and the Southern annular mode (SAM), as suggested by previous studies^{34,35}. Here we compute the spectral coherence of the AADO index with the Nino 3 index and the PQDO index. In Fig. B1d, the AADO and Nino 3 indices are strongly coherent at a spectral peak within 2–4 years, corresponding to the ENSO-sea ice teleconnection on the interannual scale. In Fig. B1e, the coherence between the PQDO and the AADO is most prominent in the frequency band of 8–16 years, indicating a decadal connection that is significantly differentiated from ENSO. Therefore, we may conclude that the influences of PQDO and ENSO on the Ross-Amundsen Seas SIC vary primarily on the temporal scales, which are in line with the spectral peaks of ENSO and PQDO on the interannual and quasi-decadal timescales, respectively.

As suggested in previous studies¹, the stationary Rossby wave trains have been recognized on a variety of temporal scales (e.g., interannual and multidecadal), connecting the tropical Pacific SST variability and Antarctic sea ice. The SST warming over the tropical Pacific excites a Rossby wave train towards the Antarctic, resulting in a significantly weakened Amundsen Sea Low (ASL) downstream, which plays a critical role in modulating the sea ice via local dynamic/thermodynamic process. In Supplementary Fig. 5 in the manuscript, the ASL exhibits prominent spectral peaks at 12–15 years that are synchronized to the PQDO and the AADO. Nevertheless, the ASL also shows periodicity on the interannual scale, which agrees with the ENSO-related

teleconnection. Thus, we may conclude that, despite the proximity in the tropical-polar wave train, the PQDO-related teleconnection pattern is most significant on decadal scales and can be well separated from the ENSO due to different timescales.

Based on the above analysis, we may conclude that the PQDO has significant decadal variability, while ENSO is most prominent on the interannual scale so that they can be separated via lowpass filtering. The decadal oscillation in the Ross-Amundsen SIC is likely driven by the PQDO since they are closely correlated and share a consistent quasi-periodicity of 8–16 years. Despite the proximity in the tropical-polar teleconnection pattern, the weakened ASL downstream of the Rossby wave train also shows considerable variation on the frequency band of 8–16 years, which cannot be explained by the ENSO signal. The PQDO-AADO relationship highlights the decadal connection between the tropical Pacific and the Antarctic, which is distinguishable from the ENSO-related teleconnections on the interannual scale. In this study, we employ an 8-yr lowpass filter on the data so that the influence of the PQDO on Antarctic sea ice could be isolated on the decadal scales while excluding the interannual ENSO signal to some extent. (please see lines 297-304 and 182-185)

Fig. B1 The power spectrums of (a) the Niño 3 index, (b) the PQDO index, and (c) the AADO index; The spectral coherence of the AADO with (d) the Niño 3 index and (e) the PQDO index.

2. Finally, the paper is overall well written and presented, but there is sometimes a lack of precision or inadequate wording. The paper would therefore benefit from careful reading and editing. The figures are well chosen and properly contribute to the argumentation, but their overall quality can be improved (size of labels and legends, missing units, map orientation, caption ...).

Reply: We appreciate the reviewer's helpful comments, which significantly improve the quality of our manuscript. The inaccurate statements and the details of figures are carefully revised throughout the manuscript following the reviewer's suggestions. Please see the detailed comments below.

Detailed comments

L22: *“Over the satellite era 1981-2020, Antarctic sea ice exhibits a substantial increasing trend [...]”. I wouldn't consider the Antarctic sea ice extent trend as “substantial”, see for instance Yuan et al. (2017). In any case, it contradicts the statement made at line 24: “[...] the dramatic decline of Antarctic sea ice in 2014-2018 [...]”*

Reply: Thank you. We've made changes to this inaccurate statement.

L28 (and elsewhere in the manuscript): *“anticorrelated” would be more precise than “out of phase”.*

Reply: Thank you. It has been revised throughout the manuscript.

L54: *“intermediate climate models” → unclear*

Reply: Thank you. A previous study evaluated the predictability of Antarctic sea ice on the interannual scale using the LOVECLIM1.2 Earth system model, which consists of atmospheric, oceanic, and vegetation modules. The LOVECLIM is an intermediate complexity model in comparison with the more sophisticated, fully coupled model in the Coupled Model Intercomparison Project (CMIP).

L60: *“The low-frequency oscillatory phenomena have [...]”* → *“Low-frequency oscillatory phenomena have [...]”*

Reply: Thank you. It has been revised.

L62: *“interdecadal~multidecadal”* → *“inter- to pluri-decadal”*

Reply: Thank you. We’ve made changes to this point.

L64: *“One of which”* → *“One of these decadal modes”*

Reply: Thank you. It has been revised.

L65: *“warming and cooling phases over time”* → *“warming and cooling phases”*.

Reply: Thank you. It has been revised.

L66: *“referring”* → *“referred”*

Reply: Thank you. It has been revised.

L72: *“fundamentally different from the dynamic nature of ENSO”* → *“fundamentally different from ENSO”*.

Reply: Thank you. This inaccurate statement has been changed.

L80: *“The underlying mechanism is inspected based on the pacemaker experiment using the ICTPAGCM-NEMO coupled model implemented with a dynamic sea ice module and historical simulations from CMIP6”* → *“The underlying mechanism is inspected in historical CMIP6 simulations and a dedicated pacemaker experiment conducted with the ICTPAGCM-NEMO coupled model”*.

Reply: Thank you for your comments. The statement is revised following the reviewer’s suggestion.

L87: *what is the dataset used here?*

Reply: Thank you for pointing this out. The sea ice concentration is derived from the

NSIDC dataset, which is introduced in Data and Method section. The dataset will be cited here in the revised manuscript.

L89: “The corresponding PC1 shows an overall icing trend” → “an overall increase in sea ice cover”. Can the positive trend of PC1 be directly translated into an increase in sea ice cover, since both positive and negative correlations with PC1 exist?

Reply: Thank you. The PC1 shows a significant trend in Antarctic SIC. Through comparing the corresponding EOF1 and the observed trend pattern (Fig. B2), the positive (negative) regression with PC1 overall agrees with the increasing (decreasing) trends of SIC in the Weddell Sea and east to the Ross Sea (the Amundsen Sea and the Bellingshausen Sea). We’ll make the statement clearer in the revised manuscript.

Fig. B2 The trend pattern of Antarctic SIC computed from 1981 to 2020 (unit: 1/yr).

L91: “The EOF2” → “The second mode of EOF”

Reply: Thank you. It has been revised.

L96: “12~14” → “12-14”

Reply: Thank you. It has been revised.

L100: “The decline of SIC from 2012 and recomb from 2018 may be a part of the current cycle” → Precise what do you mean by “SIC” here: is it the Antarctic sea ice

extent? sea ice concentration in the Ross Sea? The AADO index?

Reply: Thank you. The “SIC” described here is revealed by the PC2 of Antarctic sea ice concentration, which shows consistent decadal fluctuations as in the Ross-Amundsen Seas SIC. We’ll make the statement clearer in the revised manuscript.

L103: “Antarctic sea ice” → “Antarctic sea ice variability”.

Reply: Thank you. It has been revised.

L105: it would be valuable to state here how you define the AADO index (as stated in the “Index definitions” part.

Reply: Thank you for the reviewer’s suggestion. The definition of the AADO index is stated here in the revised manuscript.

L109: “thermodynamic/dynamics” → “thermodynamical and dynamical”

Reply: Thank you. We’ve made changes to this point.

L109: “associated with the changes in large-scale atmospheric circulations” → “largely driven by large-scale atmospheric circulation”?

Reply: Thank you. It has been revised.

L112: “AADO strongly coincides with local surface air temperature (SAT)” → as SAT is also strongly responding to SIC changes, SAT warming could be explained by a decrease in SIC induced by other factors. SAT warming still coincides with warm advection and SW radiations though.

Reply: We would like to thank the reviewer’s helpful comments. As shown in Fig. 2c (in the manuscript), surface air temperature (SAT) strongly correlates with SIC ($r = -0.97$), exhibiting a synchronized quasi-periodic variation. Here we further calculate the lead-lag correlation to identify the causality between SAT and SIC. In Fig. B3, the distribution of lead-lag correlation coefficients deviates to the left, with greater coherence when SAT leads than lags, SIC by 1 to 3 months. In other words, the SIC

melting driven by preceding and contemporary SAT warming is stronger than its feedback to maintain the anomalous SAT. Despite the lagged response of SAT may not be fully excluded here, the SAT warming, induced by the weakened ASL, still has an important triggering effect on sea ice melting. In the revised manuscript, we will discuss the interaction between SAT and SIC. Please see lines 125-129.

Fig. B3 The lead-lag correlation between the AADO index and the SAT index, which is calculated from the monthly data. The x-axis denotes the lead months of SAT.

L124: what do you mean by “sea ice draft effect”?

Reply: Thank you. It should be sea ice drift here.

L124: “The advection term” → Advection of what? “Sea ice advection”?

Reply: Thank you. It means the advection of sea ice concentration. The advection term is calculated using the following equation ²⁴:

$$SIC_{adv} = -\left(u \cdot \frac{\partial SIC}{\partial x} + v \cdot \frac{\partial SIC}{\partial y}\right)$$

where u and v are sea ice motions, x and y denote longitudes and latitudes. The advection term is calculated based on annual mean SIC and sea ice drift data. Note that the sea ice motions in the original manuscript are estimated using surface winds, which may introduce a large bias. In the revised manuscript, the sea ice motions are replaced by observed sea ice drift derived from the NSIDC Polar Pathfinder Daily 25km EASE-

Grid Sea Ice Motion Vectors, Version 4. The description of this dataset and the role of advection on the sea ice over the Ross-Amundsen Seas is discussed in the revised manuscript. (please see lines 378-381 and 145-154)

L134: *“Antarctic sea ice is closely”* → *“Antarctic sea ice variability is closely”*

Reply: Thank you. It has been revised.

L150: *“convections”* → *“convection”*

Reply: Thank you. It has been revised.

L167: *“the center of sea ice”* → *unclear*

Reply: Thank you. The simulated SIC in response to the PQDO shifts westward compared with the observation, showing a significant decline only in the Ross Sea. We'll make the statement more accurate and discuss the model bias in the revised manuscript.

L177: *how do you define the “quasi-decadal variance ratio”*

Reply: Thank you for your comment. The quasi-decadal variance ratio is defined as the variance of 8–16-yr bandpass filtered series divided by the variance of the raw series. It is used to evaluate the relative importance of the decadal component of SST/SIC simulated in the selected CMIP6 models.

L179: *“There are six of them”* → *“Six of them “*

Reply: Thank you. It has been revised.

L182: *“correlation coefficients greater than 0.4”* → *“negative correlation coefficients lower than -0.4”*.

Reply: Thank you. The statement is made more accurate in the revised manuscript.

L183: *“those models”* → *“the models”*

Reply: Thank you. It has been revised.

L183: “above average” → the multi-model mean has little value here, I would rather say that the models with the more realistic intensity of SST variance and Antarctic Ross sea ice quasi-decadal variability simulate significant correlations between the PQDO and AADO index.

Reply: We appreciate the reviewer’s suggestion. The statement has been revised.

L186: “coherence” → correlation?

Reply: Thank you. The selection of models is based on the correlation coefficient between the AADO and the PQDO indices.

L186: “Group. 1” → “Group 1”

L188: “Group. 2” → “Group 2”

Reply: Thank you. It has been revised.

L190: the pattern in Figure 5d looks a lot like ENSO. In general, how can you confidently differentiate the PQDO from other modes of variability, such as ENSO or the IPO?

Reply: Thank you. The proximity and difference between the PQDO- and ENSO-related teleconnection patterns are discussed in the major comment. The influences of the IPO and ENSO on Antarctic sea ice are also discussed in the revised manuscript: (please see lines 297-351).

L195: “For the models in Group 2, they” → “The models in Group 2 simulate”

Reply: Thank you. It has been revised.

L201: “wave trains, on the other hand, reflects“ → “wave trains could reflect”

Reply: Thank you. It has been revised.

L203: “imitate” → “reproduce” ?

Reply: Thanks. We’ve replaced “imitate” with “reproduce” which is more accurate.

L206: remove “on the other hand”

Reply: Thank you. It has been revised.

L206: “The lack of the Rossby wave train,” → “The lack of the Rossby wave train in models of group 2”

Reply: Thank you. It has been revised.

L221: “scale” → “timescale”

Reply: Thank you. It has been revised.

L230: “lead us to” → “inform us on”?

Reply: Thank you. We’ve made changes to the statement.

L241: “In recent years” → which years?

The references to sea ice increase or decrease in this paragraph are vague and somehow misleading: are you referring to the sea ice concentration in the Ross-Amundsen sea (i.e., your AADO index), or the Antarctic sea ice extent?

Reply: Thank you for pointing this out. The decline of SIC in the Ross-Amundsen Seas has slowed down since 2018. The sea ice described here refers to the SIC in the Ross-Amundsen Seas (the AADO index). We’ve rephrased this paragraph to avoid misleading.

L242: “reclimb tendency” → “reverse tendency”?

Reply: Thank you. It has been revised.

L242: “modulated by”

Reply: Thank you. It has been revised.

L247: *rephrase.*

Reply: Thank you. It has been revised.

L250: *“downstream portion, which significantly weakens” → downstream portion significantly weakening the Amundsen Sea Low.*

Reply: Thank you. It has been revised.

L253: *“suggested” → “simulated” or “reproduced”*

Reply: Thank you. It has been revised.

L259: *“the sea ice concentration will recover” → where? Precise that this is suggested by the PQDO-based prediction model.*

Reply: Thank you. We’ve made changes to this statement as the recovery of SIC is suggested by the prediction model.

L275: *remove “used in this study”*

Reply: Thank you. It has been removed from the text.

L276: *“using algorithms” → this is very vague, either be more precise or remove this sentence while keeping the reference.*

Reply: Thank you. Following the reviewer’s suggestion, “algorithms” is removed. A detailed description can be found in the reference.

L278: *“Other sea ice data are employed from” → We also employ the OISST and the HadISST sea ice concentration dataset (both at a 1° resolution) “*

Reply: Thank you. This sentence has been rephrased.

L279: *“which are overall consistent” → What do you mean by “consistent” here?*

Reply: Thank you. The AADO index calculated using the OISST and the HadISST

datasets (Supplementary Fig. 2a) is consistent with the one using the NSIDC data.

L280: “is from” → “is obtained from”

Reply: Thank you. It has been revised.

L283: why do you use fluxes from NCEP2 and atmospheric states from ERA5? Wouldn't it be more consistent to use a single dataset here?

Reply: Thank you for your comments. In Fig. B4, we compare the correlation maps of the ASL index (ERA5) with the shortwave radiation derived from ERA5 and NCEP2 reanalysis data. The correlation patterns between the two datasets are overall consistent in the Ross-Amundsen Seas, as the shortwave radiation is strongly correlated with the ASL. The weakened ASL enhances shortwave radiation, further causing SAT warming. The relationship between the ASL and the local shortwave radiation is independent of data selection. However, we must note that the NCEP2-based shortwave radiation shows comparably more consistent decadal fluctuations with the ASL and the local SAT/SIC in the Ross-Amundsen Seas. This is consistent with previous studies, demonstrating that the shortwave radiation from the ERA5 reanalysis shows a large bias in the Southern Ocean ³⁶, while the one from the NCEP2 reanalysis has better performances ³⁷. Thus, the shortwave radiation from the NCEP2 reanalysis is employed in the manuscript.

Fig. B4 The correlation maps of the ASL index with the shortwave radiation derived from (a) the ERA5 and (b) the NCEP2 reanalysis datasets, respectively. The data have been processed by an 8–16-yr bandpass filter.

L313: LIM represents both dynamical and thermodynamical sea ice processes. What is the version of LIM used in your model?

Reply: Thank you. The LIM version 3 is used in this study.

L317: “compared to state-of-the-art models” → reference needed.

Reply: Thank you for pointing this out. The reference ²⁷ is added in the revised manuscript.

L326: “tropical central pacific” → where exactly?

Reply: Thank you. We’ve changed the “tropical central Pacific” to “central tropical Pacific”, which more accurately describes the region within 10°S–10°N and 165°E–

165°W.

L339: unless I have missed something, the model data of the pacemaker experiment does not seem to be made available. The authors could indicate where to find the other data used in the study (Observations, reanalyses, CMIP6 simulations ...)

Reply: Thank you. Access to the data used in this study is added to the Data section.

Figures:

Figure 1c: labels are tough to read.

Reply: Thank you. The labels in Figure 1c and similar figures are revised.

Fig. B5 (a) The EOF2 pattern of Antarctic sea ice concentration (SIC). (b) The corresponding PC2 and the AADO index defined as the area-weighted average of SIC in the Ross-Amundsen Sea (180° – 125° W, 60° S– 75° S). (c) The local wavelet power spectrum using the Morlet wavelet, and the global wavelet/Fourier spectrums (right-hand panel) of the AADO index. The yellow contour indicates the 95% significance level using a red-noise background spectrum. The red dash line in the right-hand panel indicates the 95% confidence level for the global wavelet spectrum. (d) The 8-yr lowpass filtered series of the AADO index and the PC2 for the period 1981-2020. The gray shading indicates the positive phase of the AADO. The series has been detrended and normalized.

Figure 2a: Caption: “The correlation maps of (a) the AADO index (multiplied by -1), (b) domain-averaged surface air temperature (SAT) with 200 hPa geopotential height”
 → “The correlation maps of (a) the AADO index (multiplied by -1) and (b) surface air temperature (SAT) averaged over the Ross Sea (green box) with 200 hPa geopotential height”

Reply: Thank you. It has been revised.

Figure 3a: missing unit/label for the SST variance color bar.

Figure 3b: labels are tough to read, missing x-axis labels or units, and y-axis units.

Reply: Thank you. The problems mentioned by the reviewer have been revised.

Fig. B6 (a) The SST variance on the decadal scale (8-16-yr bandpass filtered; unit: K). (b) The cross-wavelet transform analysis between the PQDO and the AADO. The black contour indicates a 95% confidence level. The relative phase relationship is shown as arrows (with in-phase pointing right). (c) the 8-yr lowpass filtered time series of the PQDO index and the AADO index for the period 1981–2020 (detrended and normalized).

Figure 4: Missing labels on the color bar

Figure 4b: Keep consistency between the legend (AADO) and figure caption (Ross-Amundsen Sea SIC)

Figure 4c and d: using the same color bar scale would ease comparison.

Reply: Thank you. The problems mentioned by the reviewer have been revised.

Fig. B7 The correlation maps of the PQDO index with 200 hPa geopotential height (contour) and SAT (shading) in the (a) observation and (b) the CP_EXP. (c) and (d) are the correlation maps of the PQDO with Antarctic SIC in the observation and the CP_EXP, respectively, and have been preprocessed by an 8-yr lowpass filtering. (e) The detrended and normalized time series of the PQDO index and the simulated AADO index in the CP_EXP. The model output is analyzed from 1981 to 2020, consistent with

the observation.

Figure 6: what is shown on the vertical axis? the caption says “antarctic sea ice”, which is unclear, are you referring to the AADO index? Antarctic sea ice extent?

Reply: Thank you. It should be the AADO index. We’ve made a change to the caption in the revised manuscript.

Figures with maps: (Figures 1a, Fig sup. 1a, 2a, 2b, 4a, 4b, 8a, 8b, 11a, and 11b) I would recommend using the standard map orientation (lon=0) on top to make the map easier to read.

Reply: Thank you. The problems mentioned by the reviewer have been revised.

References:

Yuan et al. (2017), Increase of the Antarctic Sea Ice Extent is highly significant only in the Ross Sea, Scientific Reports.

Reviewer #3 (Remarks to the Author):

Title: Decadal oscillation provides skillful multiyear predictions of Antarctic sea ice

Authors: Liu et al.

Overview:

This manuscript focuses on the mechanisms and predictability of Antarctic sea ice on decadal timescales. The authors analyzed observational datasets and coupled climate model experiments to identify a strong link between the low-frequency variability of Antarctic sea ice and tropical Pacific variability on quasi-decadal timescales. The manuscript also presents a simple statistical model that demonstrates the predictability of this variability several years in advance.

I believe that the manuscript's findings are highly significant and valuable in advancing our understanding of Antarctic sea ice variability and improving climate predictability on longer timescales. Given the importance of Antarctic sea ice as a metric for climate change assessment and the challenges of simulating it accurately in global climate models, these insights are particularly valuable.

The manuscript is clearly and logically presented, with strong writing throughout. While I have a few suggestions that could help readers better understand the content, these do not affect the manuscript's conclusions. Therefore, I recommend a minor revision. Please find my detailed comments below.

Reply: We really appreciate the reviewer's approval of our work. We would like to thank the reviewer's supportive and insightful comments and suggestions, which significantly improved the quality of our manuscript. In light of these suggestions, we have made a substantial revision to our work, and the point-to-point responses are listed as follows:

Minor comments:

1. Line 67: "ENSO": It appears to be the first use of the acronym, and therefore it should be spelled out.

Reply: Thank you for pointing this out. It has been clarified in the manuscript.

2. Line 124: "The advection term": Please clarify which advection it is. I believe it would be "the sea-ice advection," but I am unsure whether it is the vertical, zonal, or meridional advection due to the mean flow or anomaly flow.

Reply: Thank you for your comments. "The advection term" means the horizontal advection of sea ice concentration (SIC) calculated via the following equation ²⁴:

$$SIC_{adv} = -(u \cdot \frac{\partial SIC}{\partial x} + v \cdot \frac{\partial SIC}{\partial y})$$

where u and v are sea ice motions, x and y denote longitudes and latitudes. The advection term is calculated based on monthly SIC and sea ice drift data. Note that the sea ice motions in the original manuscript are estimated using surface winds, which may introduce a large bias. In the revised manuscript, the sea ice motions are replaced by observed sea ice drift derived from the NSIDC Polar Pathfinder Daily 25km EASE-Grid Sea Ice Motion Vectors, Version 4. The description of this dataset is added to the revised manuscript (please see lines 324-329). The role of SIC advection is also discussed. (please see lines 145-154)

3. Line 161: "In the pacemaker experiment": This term appears suddenly. It would be better to refer to the method section to avoid confusing the reader.

Reply: Thank you. A reference to the method section is added here.

4. Line 209: Remove "probably."

Reply: Thank you. It has been revised.

5. Discussion section: Please include a few sentences or a paragraph to discuss climate prediction using a dynamical model. The authors analyzed fully coupled climate models (the pacemaker experiment and CMIP6 models), but the prediction model is a simple

statistical model. Initializing the sea-ice condition in the climate model is challenging due to the observational data period and quality. However, according to the results in this manuscript, the sea-ice condition in the climate model may be initialized using the tropical Pacific SST only, and such a system may demonstrate multi-year predictability. The impact of model performance on predictability could also be discussed from the perspective of CMIP6 simulations. Such discussion would enhance the manuscript's importance to a broader community.

Reply: We appreciate the reviewer's insightful comments. Following the reviewer's suggestion, we intend to discuss the implications of the PQDO-AADO teleconnection and their oscillatory features to improve the decadal predictability of Antarctic sea ice. As suggested in previous studies ²³, the decadal predictability of the Ross-Amundsen Seas sea ice remains relatively low, even with refined model initializations of local SST, SIC, subsurface temperature, and salinity. The initialization of sea ice conditions in the climate model is also challenging due to a limited observational period and data quality. In this study, the PQDO-based simple statistical model can capture the multi-year variability of sea ice well. The implication is that we highlight the importance of introducing the PQDO signal into climate model initializations when predicting Antarctic sea ice, which may be a plausible approach to further improve the multi-year predictability of sea ice in more sophisticated physical models.

The relationship between the PQDO and the AADO can be reproduced by some CMIP6 models. However, the model's performance in simulating the tropical-polar teleconnection could influence the reproducibility of the decadal oscillatory feature in the Ross-Amundsen Seas SIC, which may have further impacts on the multi-year predictability of sea ice. (please see lines 288-296)

6. Lines 317-319: Please include the horizontal resolution of the NEMO model.

Reply: Thank you. The NEMO model solves primitive equations (z-coordinate) on a tripolar ORCA2 grid (horizontal resolution of $2^{\circ}\times 2^{\circ}$, and $0.5^{\circ}\times 0.5^{\circ}$ in the tropics), which are made clear in the revised manuscript.

7. Lines 325-326: "the observed SST over the tropical Central Pacific": Please include the latitude-longitude domain for the tropical Central Pacific. The data source for the observed SST is also required. The method is also unclear: is it nudged to the observed SST, used the heat flux, or replaced with the observed SST?

Reply: Thank you. The SST data is derived from the ERSSTv5. In the pacemaker experiment, the coupled model is relaxed to the observed monthly varying SST over the central tropical Pacific (10°S–10°N and 165°E–165°W) while allowing the model to freely evolve outside the central tropical Pacific region so that it can more accurately simulate the variability that is inherently coupled to the PQDO signal. We'll make this clearer in the revised manuscript. (please see lines 391-392 and 400-403)

Reference:

- 1 Li, X. *et al.* Tropical teleconnection impacts on Antarctic climate changes. *Nature Reviews Earth & Environment* **2**, 680-698 (2021).
- 2 Ding, Q., Steig, E. J., Battisti, D. S. & Wallace, J. M. Influence of the tropics on the southern annular mode. *Journal of Climate* **25**, 6330-6348 (2012).
- 3 L'Heureux, M. L. & Thompson, D. W. Observed relationships between the El Niño–Southern Oscillation and the extratropical zonal-mean circulation. *Journal of Climate* **19**, 276-287 (2006).
- 4 Fogt, R. L. & Bromwich, D. H. Decadal variability of the ENSO teleconnection to the high-latitude South Pacific governed by coupling with the southern annular mode. *Journal of Climate* **19**, 979-997 (2006).
- 5 Meehl, G. A., Arblaster, J. M., Bitz, C. M., Chung, C. T. & Teng, H. Antarctic sea-ice expansion between 2000 and 2014 driven by tropical Pacific decadal climate variability. *Nature Geoscience* **9**, 590-595 (2016).
- 6 Clem, K. R., Renwick, J. A., McGregor, J. & Fogt, R. L. The relative influence of ENSO and SAM on Antarctic Peninsula climate. *Journal of Geophysical Research: Atmospheres* **121**, 9324-9341 (2016).
- 7 Chunhan, J., Bin, W. & Jian, L. Emerging pacific quasi-decadal oscillation over the past 70 years. *Geophysical Research Letters* **48**, e2020GL090851 (2021).
- 8 Meehl, G. A., Hu, A. & Santer, B. D. The mid-1970s climate shift in the Pacific and the relative roles of forced versus inherent decadal variability. *Journal of Climate* **22**, 780-792 (2009).
- 9 Meehl, G. A., Hu, A., Santer, B. D. & Xie, S.-P. Contribution of the Interdecadal Pacific Oscillation to twentieth-century global surface temperature trends. *Nature Climate Change* **6**, 1005-1008 (2016).
- 10 Power, S., Casey, T., Folland, C., Colman, A. & Mehta, V. Inter-decadal modulation of the impact of ENSO on Australia. *Climate dynamics* **15**, 319-324 (1999).

- 11 Jin, C., Wang, B. & Liu, J. Why Pacific quasi-decadal oscillation has emerged since the
mid-20th century. *Environmental Research Letters* **17**, 124039 (2022).
- 12 Newman, M. *et al.* The Pacific decadal oscillation, revisited. *Journal of Climate* **29**, 4399-
4427 (2016).
- 13 Newman, M., Compo, G. P. & Alexander, M. A. ENSO-forced variability of the Pacific
decadal oscillation. *Journal of Climate* **16**, 3853-3857 (2003).
- 14 Power, S. & Colman, R. Multi-year predictability in a coupled general circulation model.
Climate Dynamics **26**, 247-272 (2006).
- 15 Meehl, G. A. & Hu, A. Megadroughts in the Indian monsoon region and southwest North
America and a mechanism for associated multidecadal Pacific sea surface temperature
anomalies. *Journal of Climate* **19**, 1605-1623 (2006).
- 16 Liu, C., Zhang, W., Jin, F. F., Stuecker, M. F. & Geng, L. Equatorial Origin of the Observed
Tropical Pacific Quasi-Decadal Variability From ENSO Nonlinearity. *Geophysical Research
Letters* **49**, e2022GL097903 (2022).
- 17 Joh, Y. & Di Lorenzo, E. Interactions between Kuroshio Extension and Central Tropical
Pacific lead to preferred decadal-timescale oscillations in Pacific climate. *Scientific reports*
9, 13558 (2019).
- 18 Joh, Y., Di Lorenzo, E., Siqueira, L. & Kirtman, B. P. Enhanced interactions of Kuroshio
Extension with tropical Pacific in a changing climate. *Scientific reports* **11**, 1-12 (2021).
- 19 Di Lorenzo, E. *et al.* North Pacific Gyre Oscillation links ocean climate and ecosystem
change. *Geophysical Research Letters* **35** (2008).
- 20 North, G. R., Bell, T. L., Cahalan, R. F. & Moeng, F. J. Sampling errors in the estimation of
empirical orthogonal functions. *Monthly weather review* **110**, 699-706 (1982).
- 21 Good, S. A., Martin, M. J. & Rayner, N. A. EN4: Quality controlled ocean temperature and
salinity profiles and monthly objective analyses with uncertainty estimates. *Journal of
Geophysical Research: Oceans* **118**, 6704-6716 (2013).
- 22 Meehl, G. A. *et al.* Sustained ocean changes contributed to sudden Antarctic sea ice retreat
in late 2016. *Nature communications* **10**, 14 (2019).
- 23 Morioka, Y., Iovino, D., Cipollone, A., Masina, S. & Behera, S. K. Decadal Sea Ice Prediction
in the West Antarctic Seas with Ocean and Sea Ice Initializations. *Communications Earth
& Environment* **3**, 189 (2022).
- 24 Holmes, C. R., Holland, P. R. & Bracegirdle, T. J. Compensating biases and a noteworthy
success in the CMIP5 representation of Antarctic sea ice processes. *Geophysical Research
Letters* **46**, 4299-4307 (2019).
- 25 Armitage, T. W., Manucharyan, G. E., Petty, A. A., Kwok, R. & Thompson, A. F. Enhanced
eddy activity in the Beaufort Gyre in response to sea ice loss. *Nature Communications* **11**,
761 (2020).
- 26 Landy, J. C. *et al.* A year-round satellite sea-ice thickness record from CryoSat-2. *Nature*
609, 517-522 (2022).
- 27 Kucharski, F. *et al.* On the need of intermediate complexity general circulation models: A
“SPEEDY” example. *Bulletin of the American Meteorological Society* **94**, 25-30 (2013).
- 28 Kucharski, F. *et al.* The teleconnection of the tropical Atlantic to Indo-Pacific sea surface
temperatures on inter-annual to centennial time scales: a review of recent findings.
Atmosphere **7**, 29 (2016).

- 29 D'Arrigo, R., Cook, E. R., Wilson, R. J., Allan, R. & Mann, M. E. On the variability of ENSO over the past six centuries. *Geophysical Research Letters* **32** (2005).
- 30 Stuecker, M. F., Timmermann, A., Jin, F.-F., McGregor, S. & Ren, H.-L. A combination mode of the annual cycle and the El Niño/Southern Oscillation. *Nature Geoscience* **6**, 540-544 (2013).
- 31 Jin, F.-F. An equatorial ocean recharge paradigm for ENSO. Part I: Conceptual model. *Journal of the atmospheric sciences* **54**, 811-829 (1997).
- 32 Jin, F.-F. An equatorial ocean recharge paradigm for ENSO. Part II: A stripped-down coupled model. *Journal of the Atmospheric Sciences* **54**, 830-847 (1997).
- 33 McCreary Jr, J. P. A model of tropical ocean-atmosphere interaction. *Monthly Weather Review* **111**, 370-387 (1983).
- 34 Stammerjohn, S. E., Martinson, D., Smith, R., Yuan, X. & Rind, D. Trends in Antarctic annual sea ice retreat and advance and their relation to El Niño–Southern Oscillation and Southern Annular Mode variability. *Journal of Geophysical Research: Oceans* **113** (2008).
- 35 Yuan, X. ENSO-related impacts on Antarctic sea ice: a synthesis of phenomenon and mechanisms. *Antarctic Science* **16**, 415-425 (2004).
- 36 Mallet, M. D., Alexander, S. P., Protat, A. & Fiddes, S. L. Reducing Southern Ocean shortwave radiation errors in the ERA5 reanalysis with machine learning and 25 years of surface observations. *Artificial Intelligence for the Earth Systems* **2**, e220044 (2023).
- 37 Yu, L., Jin, X. & Schulz, E. W. Surface heat budget in the Southern Ocean from 42 S to the Antarctic marginal ice zone: four atmospheric reanalyses versus icebreaker Aurora Australis measurements. *Polar Research* (2019).

REVIEWERS' COMMENTS

Reviewer #3 (Remarks to the Author):

This is my second review. I have read the revised manuscript and found that the authors did a great job addressing all the reviewers' concerns. Therefore, I would recommend acceptance of this revised manuscript.

Reviewer #4 (Remarks to the Author):

The authors identify an Antarctic sea ice decadal oscillation (AADO) with a quasi-period of 8-16 years and show that it is linked to the Pacific Quasi-Decadal Oscillation (PQDO), which has a similar period. The authors use a variety of datasets (observations, CMIP6 historical simulations, and pacemaker climate model experiments) to show the existence of the AADO and its connection to the PQDO. Based on this connection, they introduce a simple forecast model that predicts the AADO seven years in advance. Their forecast of the development of the AADO until 2027 represents a direct test of their findings.

The authors have taken up the comments of the previous three Reviewers, which has much improved the manuscript. I have no detailed comments to add. I do not think that the author's findings are almost the same as those of previous studies that analyzed sea ice variability in connection to the Interdecadal Pacific Oscillation (IPO) (the objection raised by Reviewer 1). The authors show, e.g., that the two oscillations have different time scales and that after regressing out the IPO, a clear signal of the PQDO remains (Supplementary Fig. 12f).

The authors make a convincing case, and I recommend accepting the manuscript in its current form.

**The response to the reviewers of the manuscript
“Decadal oscillation provides skillful multiyear predictions of
Antarctic sea ice”**

Revised to *Nature Communications*

November 2023

Reviewer #3:

This is my second review. I have read the revised manuscript and found that the authors did a great job addressing all the reviewers' concerns. Therefore, I would recommend acceptance of this revised manuscript.

Reply: Thank you. We really appreciate the reviewer's approval of our revised manuscript. The reviewer's insightful comments and suggestions significantly improved the quality of our manuscript.

Reviewer #4:

The authors identify an Antarctic sea ice decadal oscillation (AADO) with a quasi-period of 8-16 years and show that it is linked to the Pacific Quasi-Decadal Oscillation (PQDO), which has a similar period.

The authors use a variety of datasets (observations, CMIP6 historical simulations, and pacemaker climate model experiments) to show the existence of the AADO and its connection to the PQDO. Based on this connection, they introduce a simple forecast model that predicts the AADO seven years in advance. Their forecast of the development of the AADO until 2027 represents a direct test of their findings.

The authors have taken up the comments of the previous three Reviewers, which has much improved the manuscript. I have no detailed comments to add. I do not think that the author's findings are almost the same as those of previous studies that analyzed sea ice variability in connection to the Interdecadal Pacific Oscillation (IPO) (the objection raised by Reviewer 1). The authors show, e.g., that the two oscillations have different time scales and that after regressing out the IPO, a clear signal of the PQDO remains (Supplementary Fig. 12f).

The authors make a convincing case, and I recommend accepting the manuscript in its current form.

Reply: The reviewer accurately summarized the key points of our manuscript. We appreciate the reviewer's supportive comments and approval of publishing this work. As noted by the reviewer, the PQDO-AADO teleconnection can be distinguished from the IPO in terms of time scales. This study demonstrates a decadal variability of Antarctic sea ice, providing insights into its multi-year predictability.